# CAR-T cell-mediated depletion of immunosuppressive tumor-associated macrophages promotes endogenous antitumor immunity and augments adoptive immunotherapy

Alba Rodriguez-Garcia [1,2], Rachel C. Lynn[1,2], Mathilde Poussin[1,2], Monika A. Eiva[1,2], Lauren C. Shaw[1,2], Roddy S. O'Connor[2], Nicholas G. Minutolo[1,2], Victoria Casado-Medrano[3], Gonzalo Lopez [4], Takami Matsuyama[5] & Daniel J. Powell Jr. [1,2✉]

The immunosuppressive tumor microenvironment (TME) represents a major barrier for effective immunotherapy. Tumor-associated macrophages (TAMs) are highly heterogeneous and plastic cell components of the TME which can either promote tumor progression (M2-like) or boost antitumor immunity (M1-like). Here, we demonstrate that a subset of TAMs that express folate receptor β (FRβ) possess an immunosuppressive M2-like profile. In syngeneic tumor mouse models, chimeric antigen receptor (CAR)-T cell-mediated selective elimination of FRβ+ TAMs in the TME results in an enrichment of pro-inflammatory monocytes, an influx of endogenous tumor-specific CD8+ T cells, delayed tumor progression, and prolonged survival. Preconditioning of the TME with FRβ-specific CAR-T cells also improves the effectiveness of tumor-directed anti-mesothelin CAR-T cells, while simultaneous co-administration of both CAR products does not. These results highlight the pro-tumor role of FRβ+ TAMs in the TME and the therapeutic implications of TAM-depleting agents as preparative adjuncts to conventional immunotherapies that directly target tumor antigens.

[1] Department of Pathology and Laboratory Medicine, Ovarian Cancer Research Center, Perelman School of Medicine, University of Pennsylvania, Philadelphia, PA, USA. [2] Center for Cellular Immunotherapies, Abramson Cancer Center, University of Pennsylvania, Philadelphia, PA, USA. [3] Department of Systems Pharmacology and Translational Therapeutics, Perelman School of Medicine, University of Pennsylvania, Philadelphia, PA, USA. [4] Department of Genetics and Genomic Sciences, Icahn School of Medicine at Mount Sinai, New York, NY, USA. [5] The Center for Advanced Biomedical Sciences and Swine Research, Kagoshima University, Kagoshima, Japan. ✉email: poda@pennmedicine.upenn.edu

The immunosuppressive tumor microenvironment (TME) represents a major barrier to effective tumor-specific T cell responses to cancer. Tumor-associated macrophages (TAMs) are major constituents of the TME[1], and are comprised of a vastly heterogeneous population of phenotypically, transcriptionally, and functionally distinct macrophages[2,3] that can support tumor growth through a variety of mechanisms, including secretion of growth factors, matrix degradation enzymes, and proangiogenic factors[4–6]. TAMs also suppress immune cells by producing an array of anti-inflammatory cytokines (e.g., interleukin-10 (IL-10)) as well as enzymes that deplete amino acids essential for T cell function (e.g., Arginase 1), or by expressing inhibitory immune checkpoint ligands (e.g., PD-L1)[4]. From a clinical perspective, TAM density correlates with poor survival in most types of solid cancer[7]. Still, in some experimental mouse models, macrophages play a pivotal role in the development of effective immunotherapy[8], and the preferential accumulation of proinflammatory M1 TAMs is associated with longer survival in patients suffering from a number of solid tumors[9–13].

Several strategies have been designed to target crucial aspects of macrophage biology in order to limit tumor progression such as regulation of macrophage recruitment with anti-CCR2, polarization with CD40 agonists or PI3Kγ inhibitors, survival with anti-CSF1R, phagocytosis with anti-CD47, and proangiogenic properties with anti-VEGF, which are currently being explored in clinical trials (reviewed in ref. [14]). However, since macrophages are an integral part of the innate immune system, some of these pan-macrophage therapeutic approaches may cause systemic toxicities and/or limit critical proinflammatory responses. Therefore, the development of clinically actionable agents and regimens that re-educate the TME by restraining protumor M2-like TAM subsets while promoting their antitumor M1-like counterparts is necessary.

A new generation of agents, including chimeric antigen receptor (CAR)-T cells, are currently being designed to reprogram the TME[15]. As an alternative to more conventional tumor-targeted CAR-T therapies that directly target cancer cells, the idea of targeting nontumor components of the TME, as an indirect approach to limit tumor progression, has gained interest. CAR-T cells that target cancer-associated fibroblasts[16–18], the extracellular matrix[19], or the tumor vasculature[20,21], serve as specific examples and have shown promising preclinical results. Nevertheless, the use of CAR-T cell technology for the selective targeting of specific TAM subsets has not been explored.

Folate receptor β (FRβ) is a glycophosphatidylinositol-anchored receptor expressed in acute myeloid leukemia cells, which has been explored as tumor antigen target for CAR-T cell therapy[22,23]. In addition to myeloid cancer cells, FRβ is expressed in noncancerous myeloid cells such as activated synovial macrophages in the arthritic joints of patients with rheumatoid arthritis (RA)[24] and in TAMs from many tumor types[25,26]. In pancreatic cancer patients, a high number of FRβ+ TAMs correlates with poor prognosis[27]. FRβ expression has been found on TAMs with an M2 phenotype[25] and short-lived agents such as recombinant immunotoxins[28,29], folate-conjugated liposomes[30,31], nanoparticles[32], or bispecific T cell engagers[33] have been developed in preclinical mouse models to target them. However, the functional relevance of FRβ+ TAMs in antitumor immunity has not been evaluated thoroughly and the impact of deep FRβ+ TAM depletion on more conventional immunotherapies is unexplored.

Here, we delineate the relative phenotype and function of FRβ+ and FRβ− TAMs and assess the therapeutic impact of targeting M2-like FRβ+ TAMs using a CAR-T cell strategy in murine models of ovarian cancer, colon cancer, and melanoma. Our results show that the subset of TAMs that express FRβ possesses a phenotypic, transcriptomic, and metabolic profile consistent with M2-like macrophages and exert functionally immunosuppressive effects on otherwise potent tumor-reactive T cells, whereas FRβ− TAMs are predominantly proinflammatory and M1-like in nature. Specific depletion of the FRβ+ subpopulation of TAMs in vivo using CAR-T cell therapy results in reprogramming of the TME, promotion of endogenous T cell-mediated immunity and control of tumor progression, and synergizes with a tumor antigen-targeted CAR-T cell therapy when applied as a preconditioning regimen. Finally, we demonstrate that CAR-T cells specific for human FRβ (hFRβ) specifically recognize and lyse primary patient-derived FRβ+ TAMs ex vivo, and illuminate a route toward clinical application.

## Results

**FRβ is expressed by TAMs in the TME.** Expression of FRβ has been demonstrated in human TAMs from various cancer types[25,26,34]. In the ID8 mouse ovarian cancer model, peritoneal TAMs are reported to express FRβ[32]. To validate this model, C57BL/6 mice were inoculated with $5 \times 10^6$ ID8 cells intraperitoneally (i.p.) and peritoneal cells were collected at indicated time points and phenotypically characterized using flow cytometry. A small fraction of peritoneal macrophages, as defined by their CD45, CD11b, and F4/80 expression (Supplementary Fig. 1a), showed positive surface expression of FRβ immediately (12.36 ± 4.51%) or 2 weeks (18.70 ± 3.47%) after tumor inoculation (Fig. 1a). When tumors were allowed to grow to the point of significant ascites formation (~8 weeks), the FRβ+ subset of TAMs was increased to 52.14 ± 11.42% of the total macrophage population (Fig. 1a), demonstrating that this is not a static population but one that increases over time. The FRβ+ TAMs subset expressed both CD11b and F4/80 markers, albeit at intermediate levels (Fig. 1b). Presence of FRβ+ TAMs was also detected in other murine solid tumor types such as B16 melanoma or MC38 colon adenocarcinoma, confirming that their accumulation in tumors is not restricted to ovarian cancer (Supplementary Fig. 1c, d). In contrast, FRβ expression was not detected in macrophages obtained from the spleen or in other cell types present in the TME such as tumor cells, myeloid derived suppressor cells, or regulatory T cells (Supplementary Fig. 2e, gating strategies are detailed in Supplementary Fig. 2a–d). In line with these data and the observation that soluble factors released by cancer cells mediate the upregulation of FRβ expression by human macrophages[25], macrophages obtained from tumor-free mice, which expressed low levels of FRβ, significantly upregulated FRβ when cultured in the presence of ID8 tumor cell supernatants for 24 h (Fig. 1c).

**FRβ+ TAMs display an M2-like profile.** FRβ expression is linked to an M2-like phenotype in human cell studies[25]. To compare the characteristics of FRβ+ and FRβ− TAMs, TAMs were collected from mice with established ID8 ascites and flow-sorted based on their FRβ expression (Supplementary Fig. 1b). Nucleus (DAPI (4′,6-diamidino-2-phenylindole)) and actin (phalloidin) staining showed marked differences in morphology between the two populations with FRβ− TAMs displaying a round shape that was more monocytic in appearance and consistent with M1 phenotype, whereas FRβ+ TAMs exhibited an elongated cell shape, consistent with M2-polarized macrophages (Fig. 2a) [35].

Gene expression analysis on sorted FRβ+ and FRβ− TAMs revealed a list of 3110 genes that were differentially expressed between both populations (false discovery rate <0.1), with 622 genes upregulated and 2488 downregulated in the FRβ-expressing subset of TAMs (Fig. 2b and Supplementary Data 1). FRβ+ TAMs displayed a transcriptional profile characteristic of M2-polarized macrophages. Among upregulated genes were the canonical

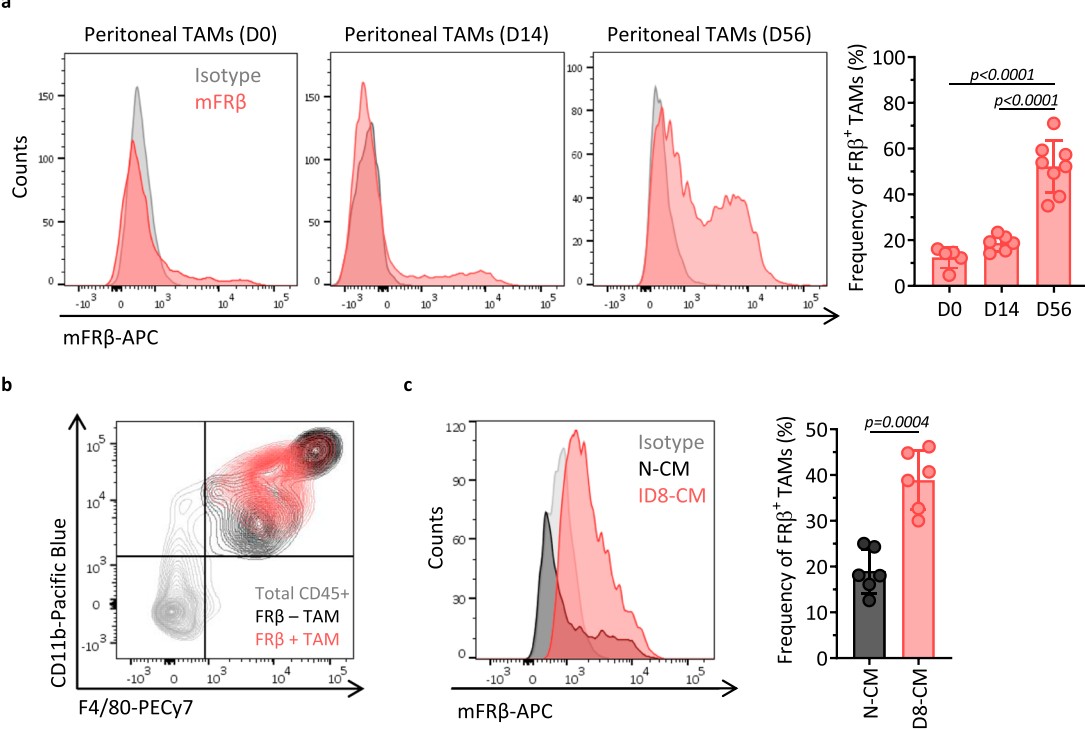

**Fig. 1 FRβ is expressed by TAMs in the ovarian cancer TME. a** mFRβ expression in peritoneal TAMs from ID8 ovarian tumor-bearing C57BL/6 mice obtained at indicated time points after i.p. tumor inoculation. Representative histograms (gray–isotype, red–anti-mFRβ, left panel) and frequency of FRβ+ TAMs (defined as live, CD45+CD11b+F4/80+ cells, right panel) are shown. Mean ± SD are represented ($n = 5$ mice for D0, 6 for D14, and 8 for D56 groups). **b** Representative flow plot demonstrating the gating strategy for TAMs by looking at CD11b (Y-axis) and F4/80 (X-axis) expression. Total CD45+ cells are depicted in gray, FRβ+ population of TAMs in red, and FRβ− TAMs in black. **c** FRβ expression in peritoneal macrophages obtained from tumor-free mice and exposed for 72 h to nonconditioned media (N-CM) or ID8-conditioned media (ID8-CM). Representative histograms (gray—isotype, black—N-CM, and red—ID8-CM, left panel) and frequency of FRβ+ macrophages (defined as live, CD45+CD11b+F4/80+ cells, right panel) are shown. Mean ± SD from two pooled independent experiments is represented ($n = 6$ mice). P values by **a** one-way ANOVA with Tukey's multiple comparison test and **c** a two-tailed paired $t$ test are indicated. Source data are provided in the Source Data file.

M2 surface markers *Msr1* (CD204), *Mrc1* (CD206), and *Cd163*, which were overexpressed by 1.52-, 2.72-, and 12.34-fold, respectively, in FRβ+ TAMs as compared to FRβ− TAMs. Genes universally associated to an M2 phenotype were also upregulated in FRβ+ TAMs, including the hallmark enzyme *Arg1* (1.65-fold), *Retnla* (or *Fizz1*, 2.23-fold), the immunosuppressive cytokine *Il10* (2.95-fold), the scavenger receptor *Stab1* (2.15-fold), the growth factor *Igf1* (1.99-fold), and genes involved in matrix remodeling such as *Adam8* (1.45-fold), *Clec7a* (1.33-fold), or *Mmp19* (1.33-fold)[36,37]. The most highly upregulated gene in FRβ+ TAMs was *Lyve1* (29.31-fold), a hyaluronan receptor, primarily expressed in lymphatic vessels but also reported in M2-like TAMs, involved in promoting lymphangiogenesis[38]. The immune checkpoint molecule *Cd274* (*Pdl1*) was also significantly upregulated in FRβ+ TAMs (1.29-fold), as well as other genes described to be expressed preferentially in M2-like macrophages such as *F13a1* (4.17-fold), *Hmox1* (2.97-fold), *Il4ra* (1.27-fold), *Trem2* (2.3-fold), *Mertk* (1.71-fold), or *Maf* (1.52-fold) (Fig. 2c)[39–46].

In contrast, known M1 gene markers such as *Gpr18* and *Serpinb2* and proinflammatory cytokines such as *Il12b*, *Il1b*, and *Il6* were significantly downregulated in FRβ+ TAMs, as compared to FRβ− TAMs (0.38-, 0.04-, 0.29-, 0.34-, and 0.63-fold, respectively) (Fig. 2b). Other M1-like genes downregulated in FRβ+ TAMs included *Ccr7* (0.44-fold), *Cd86* (0.78-fold), *Cd14* (0.44-fold), *Il27* (0.56-fold), *Cybb* (0.67-fold), and *Epas1* (*Hif2a*, 0.45-fold) (Fig. 2c)[36,39,43,47,48].

Some of the upregulated genes found at the transcriptional level were selected for further confirmation at the protein level.

Flow cytometry surface staining showed a high coexpression pattern of CD204 and CD206 with FRβ (Fig. 2d). Surface expression of PD-L1 was also higher in FRβ+ TAMs (Fig. 2e), while differential expression of other checkpoint molecules such as PD-1 or PD-L2 was not detected at either the RNA level or the protein level (Supplementary Fig. 3a). Intracellular staining also confirmed higher expression of ARG1 (Fig. 2f, left panel), as well as of the transcription factors c-MAF (*Maf*) (Supplementary Fig. S3b) and EGR2 (Fig. 2f, middle panel) in FRβ+ TAMs, the latter being proposed to be a more selective protein marker for mouse M2 macrophages[36], although this difference was not revealed at the transcriptional level. Levels of hypoxia-inducible factors (HIFs) were also assessed at the protein level in both populations. In line with the transcriptomic data, protein levels of HIF-1α were similar in FRβ+ and FRβ− TAMs, while the frequency of HIF-2α-positive cells was significantly higher in FRβ− TAMs (Supplementary Fig. 3c, d)

A multiplex cytokine assay performed on supernatants of ex vivo flow-sorted FRβ+ or FRβ− TAMs that were cultured overnight in the presence or absence of lipopolysaccharide (LPS) stimulation was used to validate the cytokine gene expression data. Higher secretion of IL-6, IL-27, IL-17α, tumor necrosis factor-α (TNF-α), interferon-β (IFN-β), and granulocyte–macrophage CSF was observed by FRβ− TAMs, while IL-10 was secreted at significantly higher levels by FRβ+ TAMs (Fig. 2g). These differences were even more pronounced for most of the analytes after stimulation with LPS. For both TAM subsets, baseline levels of cytokines such as IL-1α, IL-1β, IL-12p70, or IL-23 were below the level of detection in the absence

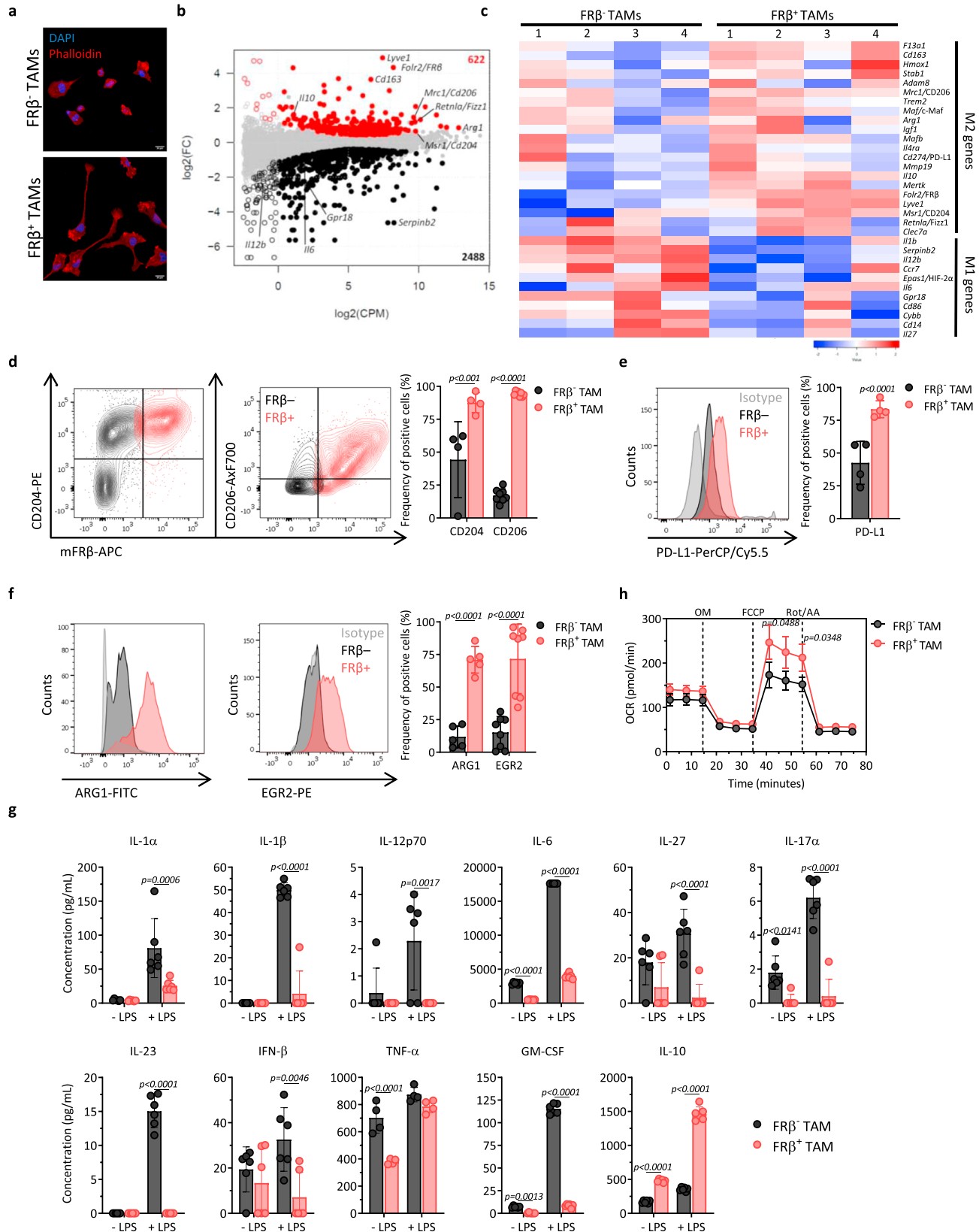

of LPS stimulation[12]. However, each of these proinflammatory cytokines were upregulated after LPS stimulation only in the FRβ− subset of TAMs (Fig. 2g).

M1- and M2-polarized macrophages also display unique metabolic properties, with M1 macrophages utilizing glycolysis to generate ATP, and M2 macrophages obtaining much of their energy from oxidative phosphorylation[49]. FRβ− and FRβ+ TAMs were harvested and directly sorted to characterize their metabolic activity using a standard Seahorse assay. Oxygen consumption rates (OCRs) were similar in FRβ− and FRβ+ TAMs at baseline,

**Fig. 2 FRβ+ TAMs display an M2-like profile.** TAMs obtained from ID8 ascites were defined as live, CD45+CD11b+F4/80+. **a** Immunofluorescence of flow-sorted FRβ− (upper panel) and FRβ+ (lower panel) TAMs stained for actin with phalloidin-rhodamine (red) and for cell nuclei with DAPI (blue). Scale bar, 10 μm. Representative images from one experiment (n = 3 mice). **b** Differential gene expression profile between FRβ− and FRβ+ TAMs as measured by RNA-seq (n = 4 per group). Log 2 of fold change (FC) vs. log 2 of counts per million (CPM) is represented. Red dots represent genes upregulated in FRβ+ TAMs, black dots represent downregulated genes, gray dots represent not differentially expressed genes, and open dots represent genes expressed at low levels. Relevant genes are highlighted. Statistics were calculated by EdgeR (cutoff:log 2(FC) = ±0.26, FDR < 0.1). **c** Heatmap of normalized mean-centered mRNA expression of genes associated to M1 or M2 macrophages in flow-sorted FRβ− and FRβ+ TAMs (n = 4 per group). **d** Representative flow plots showing coexpression of mFRβ (X-axis) and CD204 (left panel) or CD206 (middle panel, Y-axis) in FRβ+ TAMs (red) and FRβ− TAMs (black). Mean frequencies of CD204+ (n = 4 mice) and CD206+ (n = 8 mice) cells ± SD are shown (right panel). Expression of **e** PD-L1 (n = 4 mice) or **f** Arg1 (n = 5 mice) and Egr2 (n = 8 mice) in FRβ− and FRβ+ TAMs. Representative histograms (gray–isotype, black—FRβ− TAMs, red—FRβ+ TAMs, left panel) and mean frequencies (right panel) ± SD are shown. **g** Production of cytokines by flow-sorted FRβ− and FRβ+ TAMs as quantified by a cytokine bead-based assay after 24 h of incubation unstimulated or in the presence of 5 ng/mL of LPS. Mean concentrations of six replicates ± SD are represented. **h** Oxygen consumption rates (OCR) of flow-sorted FRβ− and FRβ+ TAMs after one day in culture in response to mitochondrial inhibitors added at indicated times as detected by a Seahorse assay. Mean OCR of six replicates ± SD is represented. Data are representative of **g** two and **h** three independent experiments. P values by a **d**–**h** two-way ANOVA with Sidak's multiple comparison test are indicated. OM oligomycin, FCCP phenylhydrazone, Rot rotenone, AA antimycin. Source data are provided in the Source Data file.

but under depolarizing conditions, the oxidative features of FRβ+ TAMs were significantly higher (Fig. 2h and Supplementary Fig. 4b), consistent with an increased energy-generating capacity. Of note, these higher maximal OCR levels are consistent with an M2 phenotype of oxidative metabolism. When FRβ+ TAMs were cultured in the presence of LPS, maximum OCR was reduced to similar levels as M1-like FRβ− TAMs, which were unaffected by LPS addition (Supplementary Fig. 3c), while the addition of IL-4, typically used to polarize macrophages towards an M2 phenotype[50], had the opposite effect, increasing maximum OCR in both populations (Supplementary Fig. 3d). To provide increased insight into the metabolic features of FRβ− and FRβ+ TAMs, we measured mitochondrial mass and mitochondrial membrane potential by flow cytometry by using MitoTracker FM and tetramethylrhodamine methyl ester, respectively. Similar levels were observed for both parameters in FRβ− and FRβ+ TAMs (Supplementary Fig. 4e, f), consistent with the baseline OCR profile. Finally, the extracellular acidification rate (ECAR) was measured to investigate the glycolytic capacity of FRβ− and FRβ+ TAMs. As seen in Supplementary Fig. 4g, no statistically significant differences in ECAR levels were reported. In aggregate, our findings imply that the enhanced metabolic capacity of FRβ+ TAMs is attributed to an increased energy reserve which is supported by an enhanced ability to consume oxygen at high rates under stimulatory conditions.

**FRβ+ TAMs inhibit antigen-specific T cell function.** In order to test the hypothesis that FRβ+ TAMs with M2-like features functionally suppress T cell responses, mouse CAR-T cells specific for human mesothelin (hMeso) were used to establish a model of antigen-specific T cell response. hMeso CAR-T cells were labeled with a cell trace dye and cocultured with ID8 cells engineered to express hMeso protein on the cell surface (ID8. hMeso; Fig. 3a) or parental ID8 cells in order to assess antigen-driven CAR-T cell division. Sorted FRβ+ or FRβ− TAMs were added at the inception of these cocultures, which were then incubated for 72 h. The proliferation of hMeso CAR-T cells was significantly reduced in the presence of FRβ+ TAMs as compared to the condition in which FRβ− TAMs were added (Fig. 3b). Furthermore, levels of IFN-γ secreted by the CAR-T cells were reduced in the presence of FRβ+ TAMs to a level similar to that observed when the CAR-T cells were cocultured with antigen-negative ID8 target cells. IFN-γ secretion was unaffected by the presence of FRβ− TAMs (Fig. 3c). Consistent with their ability to produce IL-10, the levels of this cytokine appeared to be higher in cocultures of CAR-T cells and ID8.hMeso cells in the presence of FRβ+ TAMs, implicating this as one possible mechanism of

immunosuppression (Supplementary Fig. 5a). However, blockade of IL-10 by antibodies did not rescue proliferation (Supplementary Fig. 5b) or IFN-γ secretion (Supplementary Fig. 5c) by CAR-T cells, suggesting that very likely IL-10 is not a major contributor for immunosuppression and other mechanisms might be involved.

Similarly, FRβ+ TAMs inhibited T cell proliferation and IFN-γ secretion when transgenic OT-1 T cells (TCR specific for OVA257−264 peptide) served as an independent model of antigen-specific T cell responses instead of CAR-T cells (Supplementary Fig. 5d–f). Collectively, these results show that FRβ+ TAMs are able to suppress both, T cells that recognize their specific antigen through a CAR or through their TCR, and suggest that deep and selective depletion of FRβ+ TAMs may functionally enhance endogenous antitumor immunity and curb tumor progression.

**Mouse FRβ (mFRβ) CAR-T cells efficiently target FRβ+ TAMs ex vivo.** CAR-T cell therapy has mediated complete remission of certain types of cancer targeting a number of different antigens. This led us to hypothesize that an FRβ-directed CAR approach will result in efficient and selective M2-like TAM depletion. A mFRβ-specific CAR-T cell platform was developed by cloning a rat anti-mFRβ scFv, derived from the CL10 antibody[29], into a previously validated MSGV retroviral CAR construct containing murine CD28, 41BB, and CD3ζ intracellular signaling domains (Fig. 4a)[20]. CAR constructs also contain green fluorescent protein (GFP) in trans to serve as a marker for transduced cells. A construct containing the mouse FMC63 scFv, specific for human CD19 (hCD19), was developed in parallel to serve as irrelevant target control (Fig. 4a)[51]. After αCD3/CD28 bead activation of total C57BL/6 mouse splenocytes, retroviral transduction, and expansion in the presence of mIL-2 (Supplementary Fig. 6a), cultures contained >98% CD3+ T cells (Supplementary Fig. 6b) with a ratio of 19:1 CD8+ to CD4+ T cells (Supplementary Fig. 6c). Surface CAR expression was confirmed in GFP+ cells using labeled recombinant mFRβ protein (Supplementary Fig. 6d) or antibodies that recognize the scFv portion of the CAR by flow cytometry (Fig. 4b). CAR transduction was reproducibly achieved at frequencies of 70–80%. Specific reactivity of mFRβ mouse CAR-T cells was confirmed by dose-dependent IFN-γ production in the presence of increasing concentrations of immobilized recombinant mFRβ protein (Supplementary Fig. 6e). To create a proof-of-concept model of CAR-targetable murine ovarian cancer, the ID8 ovarian cancer cell line was engineered to stably express mFRβ antigen (ID8.mFRβ; Supplementary Fig. 6f). mFRβ CAR-T cells released high levels of IFN-γ in an antigen-specific manner when cocultured with ID8.mFRβ cells, but not with

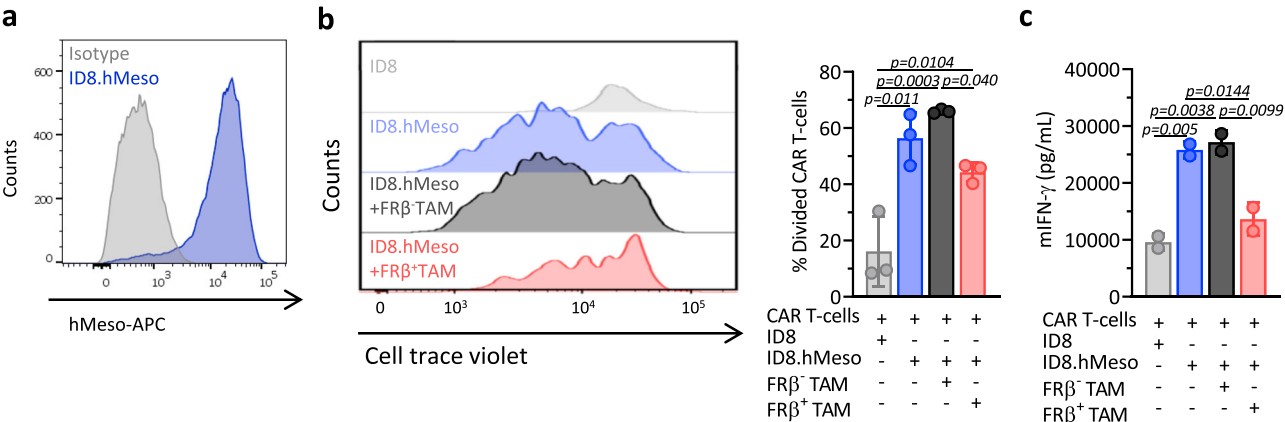

**Fig. 3 FRβ⁺ TAMs inhibit antigen-specific T cell function. a** Expression of human mesothelin (hMeso) in engineered ID8.hMeso cells as detected by flow cytometry. Representative histogram (gray–isotype, blue—hMeso) is shown. hMeso-specific mouse CAR-T cells labeled with cell trace violet (a fluorescent dye that is diluted with cell proliferation) were cocultured with ID8 or ID8.hMeso target cells in the presence of flow-sorted FRβ⁻ or FRβ⁺ TAMs for 72 h. **b** Representative histograms looking at cell trace violet staining on CAR⁺ T cells (gated on live, CD3⁺GFP⁺, left panel) as well as mean percentage of divided cells ± SD of triplicates (right panel) are shown. **c** mIFN-γ levels in 72 h supernatants as detected by ELISA. Mean of technical duplicates ± SD is shown. **b**, **c** Representative data from two independent experiments. *P* values by a one-way ANOVA with Tukey's multiple comparison test are indicated. Source data are provided in the Source Data file.

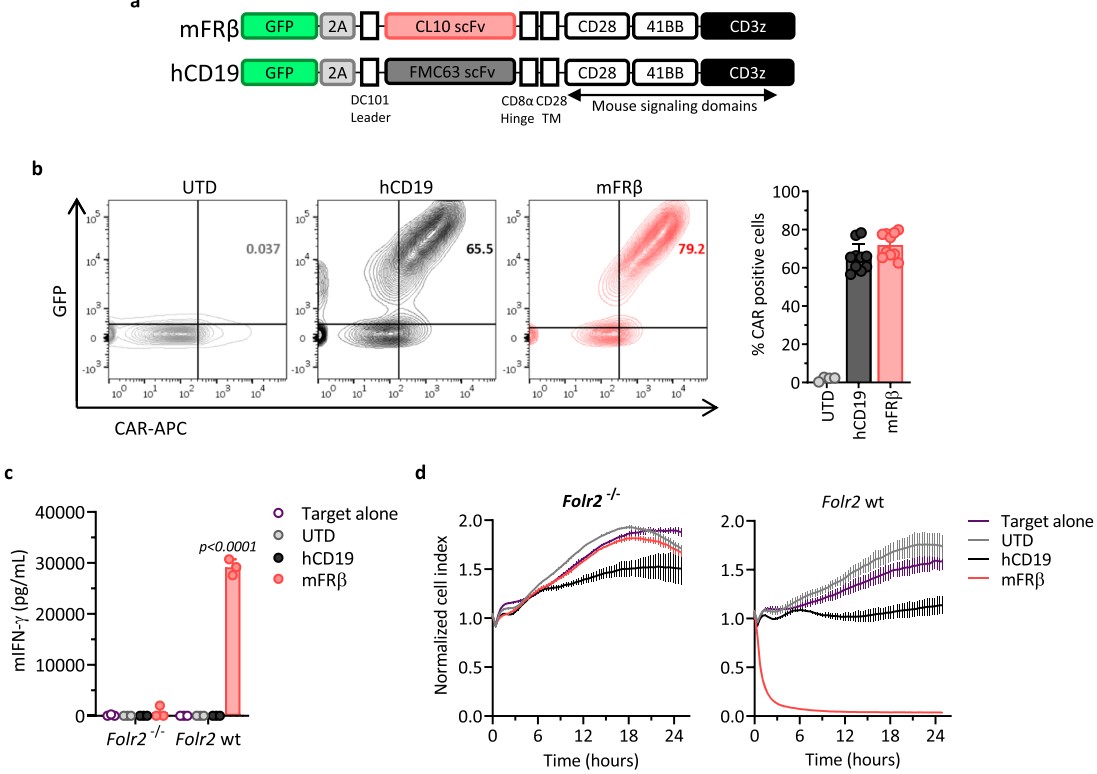

**Fig. 4 Mouse FRβ CAR-T cells efficiently target FRβ⁺ TAMs ex vivo. a** Schematic representation of retroviral CAR constructs containing the rat anti-mouse FRβ (mFRβ) scFv (CL10) or the mouse anti-human CD19 (hCD19) scFv (FMC63), linked to the mouse intracellular signaling domains CD28, 41BB, and CD3ζ in tandem. Both constructs also include GFP as a transduction marker. **b** Expression of hCD19 and mFRβ CARs in the surface of transduced mouse T cells as detected by staining with biotinylated goat anti-mouse IgG (H + L) or rabbit anti-human IgG (H + L), respectively, followed by streptavidin (SA)-APC labeling at day 5 of T cell expansion. CAR⁺ cells were defined as double positive for GFP (*Y*-axis) and APC (*X*-axis). Representative flow plots (left panel) and mean frequencies of CAR⁺ T cells (pregated on live/CD3⁺, right panel) ± SD are shown (*n* = 10). Each symbol represents an independent T cell expansion. Cocultures of mFRβ CAR, hCD19 CAR, or UTD T cells with CD11b⁺ cells obtained from ID8 tumor-bearing *Folr2* wt or *Folr2⁻/⁻* mice were established at a 1:1 E:T ratio. **c** Levels of secreted mIFN-γ as detected by ELISA in the supernatants at 24 h. **d** Real-time cytotoxicity assays performed using xCelligence technology. **c**, **d** Mean ± SD of triplicates is represented. *P* values by a two-way ANOVA with Tukey's multiple comparison test are indicated. Source data are provided in the Source Data file.

parental ID8 cells (Supplementary Fig. 6g), and specifically lysed ID8.mFRβ target cells in a real-time cytotoxicity assay using xCelligence Technology (Supplementary Fig. 6h). In vivo, multiple doses of mFRβ mouse CAR-T cells significantly delayed tumor progression in mice implanted i.p. with ID8.mFRβ RFP-fLuc cells as measured by tumor bioluminescence (Supplementary Fig. 6i). These data confirmed high and selective antigen-driven reactivity of mFRβ CAR-T cells in both in vitro and in vivo settings where tumor cells expressed the target antigen.

To test the reactivity of the CAR-T cells against their intended TAM targets, CD11b+ cells (containing TAMs) were isolated from the ascites of ID8 tumor-bearing Folr2 (gene encoding for FRβ) wild-type (WT) (Folr2 wt) or Folr2 knockout (KO) (Folr2−/−) mice and cocultured with mFRβ CAR, control hCD19 CAR, or untransduced (UTD) T cells. mFRβ CAR-T cells produced high levels of IFN-γ (Fig. 4c) and induced target cell cytolysis (Fig. 4d) only when cocultured with TAMs from Folr2 wt mice, but not TAMs from Folr2−/− mice, while control T cells showed no specific reactivity to these targets.

**Depletion of FRβ+ TAMs by mFRβ CAR-T cells in vivo delays tumor growth and reeducates the TME.** The hypothesis that elimination of FRβ+ TAMs using mFRβ CAR-T cells will inhibit tumor growth in an immunocompetent recipient was tested in the model outlined in Fig. 5a. Briefly, C57BL/6 mice bearing 21-day i.p. ID8 RFP-fLuc tumors (which lack mFRβ expression) were preconditioned with cyclophosphamide (Cy) and treated with a single dose of the different CAR+ or UTD T cells, and subsequent doses of IL-2. Preconditioning regimens with Cy and IL-2 have been widely used in both preclinical mouse models and clinical trials in order to improve engraftment, persistence, and functional activity of adoptively transferred T cells[16,20,52–55]. Animals were either euthanized at indicated time points to collect tumor ascites and analyze immune cell infiltrates by flow cytometry, or longitudinally monitored for tumor progression by bioluminescence imaging (BLI). Administration of mFRβ CAR-T cells induced transient reduction in body weight that was rapidly resolved without further signs of toxicity (Supplementary Fig. 7b). Notably, treatment with mFRβ CAR-T cells led to the specific depletion of the FRβ+ TAM population, while control hCD19 CAR or UTD T cells had no effect on macrophages at day 6 after T cell infusion (Fig. 5b). At the same time point, significant percentages of CD8+ CAR+ T cells were detected in the ascites of mice treated with GFP+ mFRβ CAR-T cells, which, in some cases, represented up to 40% of total CD8+ T cells. In contrast, GFP+ T cells were not detected in mice treated with GFP+ hCD19 CAR or UTD T cells (Supplementary Fig. 7e). Furthermore, depletion of FRβ+ TAMs by mFRβ CAR-T cells resulted in a reproducible and statistically significant delay in tumor growth as measured by BLI, in spite of the fact that the tumor cells lack mFRβ target antigen expression (Fig. 5c). This also led to a significant increase in the survival of mice treated with mFRβ CAR-T cells, relative to mice treated with UTD or hCD19 CAR-T cells (Fig. 5d). Of note, antitumor efficacy of mFRβ CAR-T cells was dependent on antigen expression (Supplementary Fig. 7a) and Cy + IL-2 preconditioning regimen (Supplementary Fig. 7c, d).

In an attempt to elucidate the mechanism behind this antitumor efficacy, the impact of FRβ+ TAM depletion on different immune cell populations was explored in the ID8 model.

Treatment with mFRβ CAR-T cells resulted in an increased recruitment of precursor myeloid cells to the TME, as evidenced by a significant elevation in the circulating number of CD11b+ cells in the blood (Fig. 5e) and higher frequencies in the ascites (Fig. 5f). The composition of the myeloid compartment in the ascites was dramatically altered after mFRβ CAR-T cell therapy. Three different

subsets were distinguished within the CD11b+ population based on Ly6C and Ly6G expression (Supplementary Fig. 7f). Frequencies of mature resident monocytes (Ly6C−Ly6G−) were decreased in mFRβ CAR-T cell treated mice, as compared to mice treated with UTD or hCD19 CAR-T cells (20 ± 6.8 vs. 61 ± 7% and 59.3 ± 7.3%, respectively), corresponding with depletion of FRβ+ TAMs (Fig. 5g). In contrast, subsets of inflammatory monocytes (Ly6C+Ly6G−) and neutrophils (Ly6CintLy6G+) were highly increased in mFRβ CAR-treated group (31.9 ± 17 and 30 ± 12.3%, respectively) as compared to UTD (8.3 ± 1.6 and 11.8 ± 7.4%) or hCD19 (9.4 ± 2.2 and 9.5 ± 3.7%) groups (Fig. 5g). Similar trends in the concentration of myeloid cell populations were observed in blood in the aftermath of mFRβ CAR-T cell treatment (Supplementary Fig. 7g). Consistently, further phenotypic analysis of infiltrating monocyte subsets revealed higher frequencies of classical/inflammatory monocytes (CX3CR1−CCR2+), while lower frequencies of non-classical/patrolling monocytes (CX3CR1+CCR2−; Supplementary Fig. 7h)[56]. Further, infiltrating CD11b+ myeloid cells isolated from the ascites of mice treated with mFRβ CAR-T cells produced lower levels of IL-10 as well as higher levels of TNF-α as compared to cells from mice treated with control UTD or hCD19 CAR-T cells (Fig. 5h). These results suggest a reeducation of the TME towards a proinflammatory one promoted by mFRβ CAR-T cell therapy.

Changes in the T cell compartment were also observed. Significantly higher numbers and frequencies of CD8+ T cells were detected in the blood (Fig. 5i) and ascites (Fig. 5j) of mFRβ CAR-treated mice, as compared to hCD19- or UTD-treated mice. Endogenous CD8+ T cells in the ascites of treated mice displayed a more activated phenotype as assessed by the upregulation of PD-1, IFN-γ, and TNF-α, compared to control groups (Fig. 5k). This was suggestive of recruitment of endogenous T cells to the tumor sites, as a contributing factor to the antitumor activity mediated by mFRβ CAR-T cells, which do not directly target tumor cells. To test this hypothesis, similar in vivo experiments as depicted in Fig. 5a were conducted first using ID8.OVA tumor cells as target cells. Following treatment, a significant increase in circulating OVA-specific CD8+ T cells was detected in the blood of mice administered mFRβ CAR-T cells (54.04 cells/μL), as compared to UTD (11.20 cells/μL) or control hCD19 CAR-T cells (17.26 cells/μL) (Fig. 5l). Higher frequencies of OVA-specific CD8+ T cells were also detected in splenocytes (3.67-fold vs. UTD and hCD19) (Fig. 5m, left panel) and ascites (2.24- and 2.43-fold vs. UTD and hCD19, respectively) (Fig. 5m, right panel) of mFRβ CAR-T cell treated mice, albeit differences did not reach statistical significance in the latter case. Next, the instrumental contribution of endogenous CD8+ T cells to antitumor activity was further corroborated by the finding that mFRβ CAR-T cells were not able to control parental ID8 tumor progression in mice that lack functional CD8+ T cells (Fig. 5n), in spite of the efficient depletion of FRβ+ TAMs (Supplementary Fig. 6h).

To investigate whether this therapeutic strategy is translatable to other tumor types, similar in vivo studies were conducted in mice bearing B16 melanoma (Supplementary Fig. 7a–d) or MC38 colon adenocarcinoma (Supplementary Fig. 7e–h) subcutaneous tumors. In both solid tumor models, mFRβ CAR-T cells depleted FRβ+ TAMs, increased T cell infiltration in tumors, and mediated significant control of tumor progression.

**Preconditioning the TME with anti-TAM CAR-T cells improves the efficacy of tumor-specific CAR-T cells.** Given the effects of the depletion of FRβ+ TAMs on endogenous antitumor immune responses, we hypothesized that the combination of mFRβ CAR-T cells with a conventional tumor-targeted CAR might result in enhanced antitumor efficacy. For that, C57BL/6 mice were inoculated with ID8.hMeso tumor cells and, after 21 days, randomized into groups for treatment with a single dose

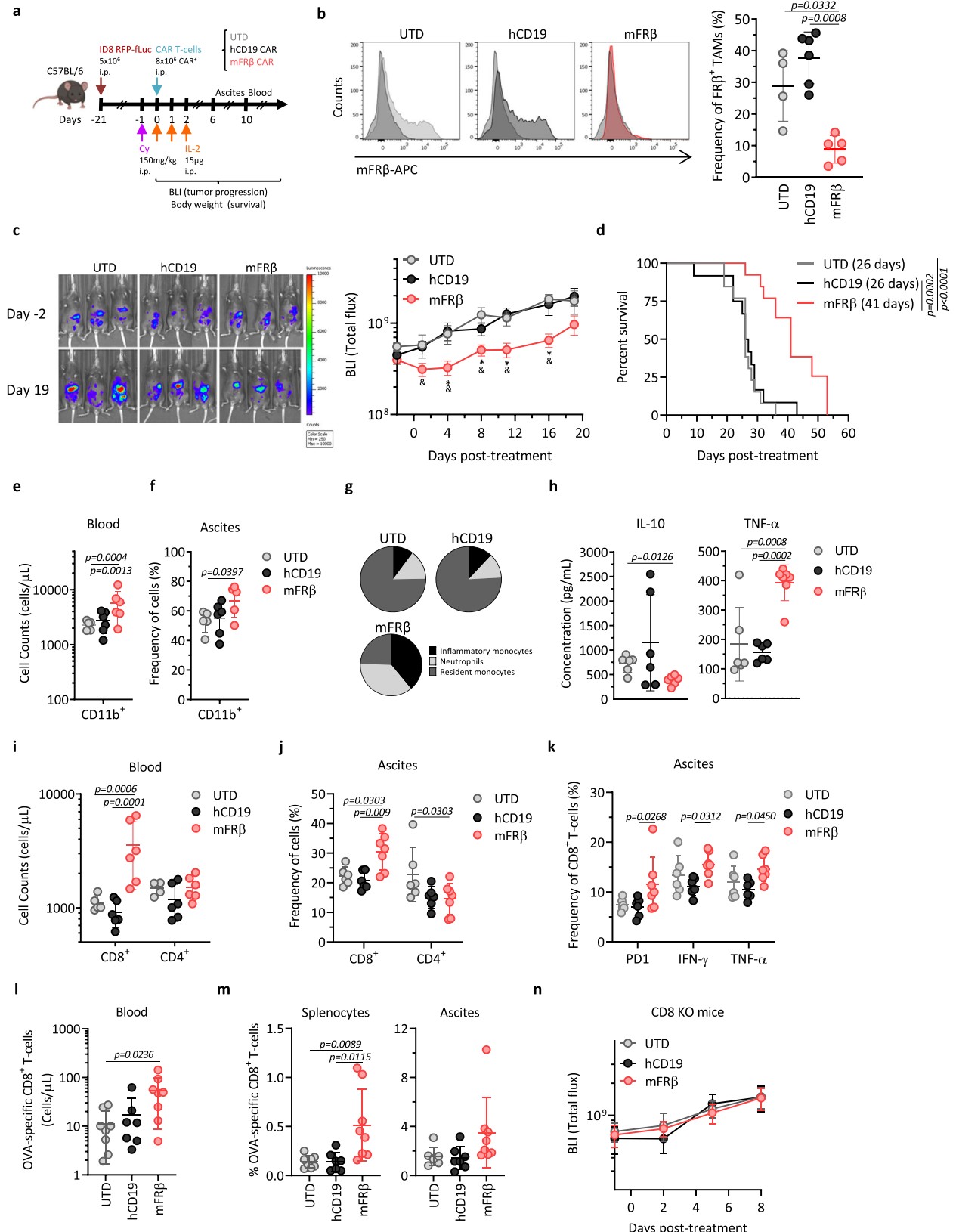

of hCD19, mFRβ, or hMeso-specific CAR-T cells (Supplementary Fig. 8a, b), or the combination of both mFRβ- and hMeso-specific CAR-T cells given simultaneously or sequentially as depicted in Fig. 6a, after Cy preconditioning. In these studies, a suboptimal T cell dosing regimen was used in order to reveal a potential

therapeutic window (Supplementary Fig. 8c). Under these conditions, the simultaneous combination of both CARs did not significantly improve antitumor efficacy (Fig. 6b, c) or survival (Fig. 6e) over either treatment alone. By contrast, mice that received the combined, staggered treatment regimen of hMeso

**Fig. 5 Depletion of FRβ+ TAMs by mouse FRβ CAR-T cells in vivo delays tumor growth and reeducates the TME. a** Schematic representation of the in vivo study. **b** mFRβ expression in peritoneal TAMs 6 days after treatment. Representative histograms (left panel) and frequency of FRβ+ TAMs (pregated as live, CD45+CD11b+F4/80+, right panel) are shown ($n = 4$ mice for UTD, 6 for hCD19, and 5 for mFRβ). **c** Tumor progression as monitored by bioluminescence imaging (BLI). Representative images (left panel) and mean radiance (right panel) are shown. **d** Kaplan–Meier survival curves. Mean survival is indicated in the legend. **c**, **d** Data pooled from three independent experiments ($n = 13$ mice for UTD and mFRβ and 12 for hCD19). Concentration of **e** CD11b+ or **i** CD8+ and CD4+ T cells in blood at day 10 ($n = 5$ mice for UTD, and 6 for hCD19 and mFRβ). Frequencies of **f** CD11b+ cells (pregated on live, CD45+), **j** CD8+ and CD4+ T cells (pregated on live, CD45+CD3+), or **k** CD8+PD-1+, CD8+IFN-γ+ and CD8+TNF-α+ T cells (pregated on live, CD45+CD3+GFP−), in the ascites of mice at day 6 ($n = 6$ mice for UTD and hCD19 and 7 for mFRβ). **g** Pie charts representing the distribution of myeloid cell subsets (pregated on live, CD45+CD11b+): inflammatory monocytes (Ly6C+Ly6G−), neutrophils (Ly6CintLy6G+), and resident monocytes (Ly6C−Ly6G−) in ascites at day 6 ($n = 5$ mice for UTD and mFRβ and 6 for hCD19). **h** Production of IL-10 (left panel) and TNF-α (right panel) by peritoneal CD11b+ cells at day 6. Concentrations were determined by a cytokine bead-based assay after 48 h of incubation in the presence of 5 ng/mL of LPS ($n = 6$ mice per group). **l**, **m** The experiment depicted in panel (**a**) was performed using ID8.OVA tumor cells. **l** Concentration of CD8+ OVA-specific T cells in blood at day 8. **m** Frequency of CD8+ OVA-specific T cells (pregated on live, CD45+CD3+) in splenocytes (left panel) and ascites (right panel) at day 8 ($n = 8$ mice for UTD and mFRβ and 7 for hCD19). **n** The experiment depicted in panel (**a**) was performed in CD8 KO mice. Tumor progression was monitored by BLI ($n = 6$ mice for UTD and hCD19, and 7 for mFRβ groups). Data are represented as mean ± SD in panels (**b**, **e**, **f**, **h-m**) or mean ± SEM in panels (**c**, **n**). **c** P values calculated by a multiple two-tailed t test (mFRβ as compared to (*) the UTD group; or (&) hCD19) and exact values are indicated as follow: D1 & = 0.0363; D4 * = 0.0264, & = 0.0131; D8 * = 0.0080, & = 0.0274; D11 * = 0.0109, & = 0.0022; D16 * = 0.0001, & = 0.0244. P values by a (**d**) log-rank Mantel–Cox test; a **b**, **e f**, **i**, **j**, **k** two-way ANOVA with Tukey's multiple comparison test; or a **h**, **l**, **m** one-way ANOVA with Tukey's multiple comparison test are indicated. Data shown are representative from **n** one or **e–m** two independent experiments. Source data are provided in the Source Data file.

CAR-T cells given 3 days after mFRβ CAR-T cells displayed a significantly improved and lasting control of tumor progression as compared to either treatment alone or to the simultaneous combination (Fig. 6b, c). In fact, this treatment resulted in tumor regressions in all of the treated mice (Fig. 6d) and extended survival (Fig. 6e).

To better understand the mechanisms of improved antitumor efficacy, CD45.1 mice were used as donors for hMeso CAR-T cells in order to be able to track them and differentiate them from mFRβ CAR-T cells, which are CD45.2. Higher concentrations of hMeso CAR-T cells were detected 9 days after hMeso CAR-T cell administration in the blood of mice that had been preconditioned with TAM-depleting mFRβ CAR-T cells (Fig. 6f), as well as increased numbers of circulating endogenous CD45.2 T cells (Fig. 6g), that were contracted within 3 weeks. In addition, an analysis of the TME at the endpoint demonstrated lower frequencies of immunosuppressive cell populations (Supplementary Fig. 8f–h) in the sequential treatment group, and low but detectable levels of both CAR-T cells (Supplementary Fig. 8d e), demonstrating their ability to persist in a favorable TME.

Similar results in terms of antitumor activity were observed in a different model system using a CAR specific for mouse mesothelin protein (Supplementary Fig. 8e–h), which is endogenously expressed by ID8 tumor cells (Supplementary Fig. 8d). These results illustrate the impact of TME preconditioning by FRβ CAR-T cells after Cy chemotherapy on the antitumor activity of tumor-specific CAR-T cells in immunocompetent mouse tumor models.

**Primary human TAMs express FRβ and are targets of hFRβ CAR-T cells.** As a route towards future clinical application, the presence of FRβ+ TAMs was assessed in macrophages obtained from the ascites of ovarian cancer patients. High frequencies of FRβ+ TAMs, as defined by the coexpression of CD11b and CD14, were detected in all of the biospecimens tested (Fig. 7a). Similar to our findings in mice, FRβ was coexpressed with the human M2 macrophage markers CD206 (Fig. 7b, left panel) and CD163 (Fig. 7b, right panel) on patient TAMs. As the development of hFRβ-specific CAR-T cells for the treatment of AML has been previously reported by our group[23], their capacity to recognize and kill primary TAMs derived from patients ex vivo was assessed. CD11b+ cells were isolated from human ascites by using magnetic microbeads and cocultured with hFRβ-specific CAR-T cells. Antigen-specific IFN-γ production (Fig. 7c) as well as

specific cytolysis of CD11b+ cells (containing the FRβ+ TAMs) was observed (Fig. 7d) from hFRβ-specific CAR-T cells, but not control T cells, establishing the feasibility of targeting hFRβ+ TAMs using human CAR-T cells.

Finally, in order to explore the clinical significance of FRβ in ovarian cancer patients, we investigated the correlation between gene expression and progression-free survival (PFS) by using the Kaplan–Meier plotter database[57]. Interestingly, while no significant correlation was observed in ovarian cancer patients at early stages of the disease (I and II), patients at stages III and IV with high FOLR2 expression had a significantly shorter median survival as compared with patients whose tumors had low expression of FOLR2 (Fig. 7e). A more extensive analysis of the correlation of FOLR2 and survival including different tumor types was performed revealing significant prognostic value in several tumor types including gastric cancer, bladder carcinoma, esophageal squamous cell carcinoma, kidney renal clear cell carcinoma, lung squamous cell carcinoma, stomach adenocarcinoma, and thymoma (Supplementary Fig. 10a, b)[58]. These results support the potential benefit of targeting FRβ-expressing immunosuppressive macrophages in cancer patients.

**Discussion**
The knowledge that TAMs play a key role in the promotion of tumor growth and the inhibition of antitumor immune responses has fostered the development and clinical testing of various agents to target them[5]. Early clinical results offer some promise of improved cancer outcomes but rates of response have generally been low[14]. Nevertheless, TAM populations are highly heterogeneous, and certain subsets have biological functions including cytotoxicity and antigen presentation, which are vital not only in the defense against pathogens but also at tumor sites[59]. Here, we hypothesized that specific depletion of the immunosuppressive subset of TAMs using CAR technology will reprogram the TME to one that promotes antitumor immune responses and limits tumor progression. Development of such a strategy holds the potential to improve the efficacy of other immunotherapies including CAR-T cell therapy.

In this study, we investigated the role of FRβ as a valid target for the selective depletion of immunosuppressive M2-like subset of TAMs, while sparing proinflammatory M1-like subsets. Our data show that a subpopulation of macrophages found in the TME express FRβ, and their frequency significantly increase with tumor progression in a clinically relevant mouse model of ovarian

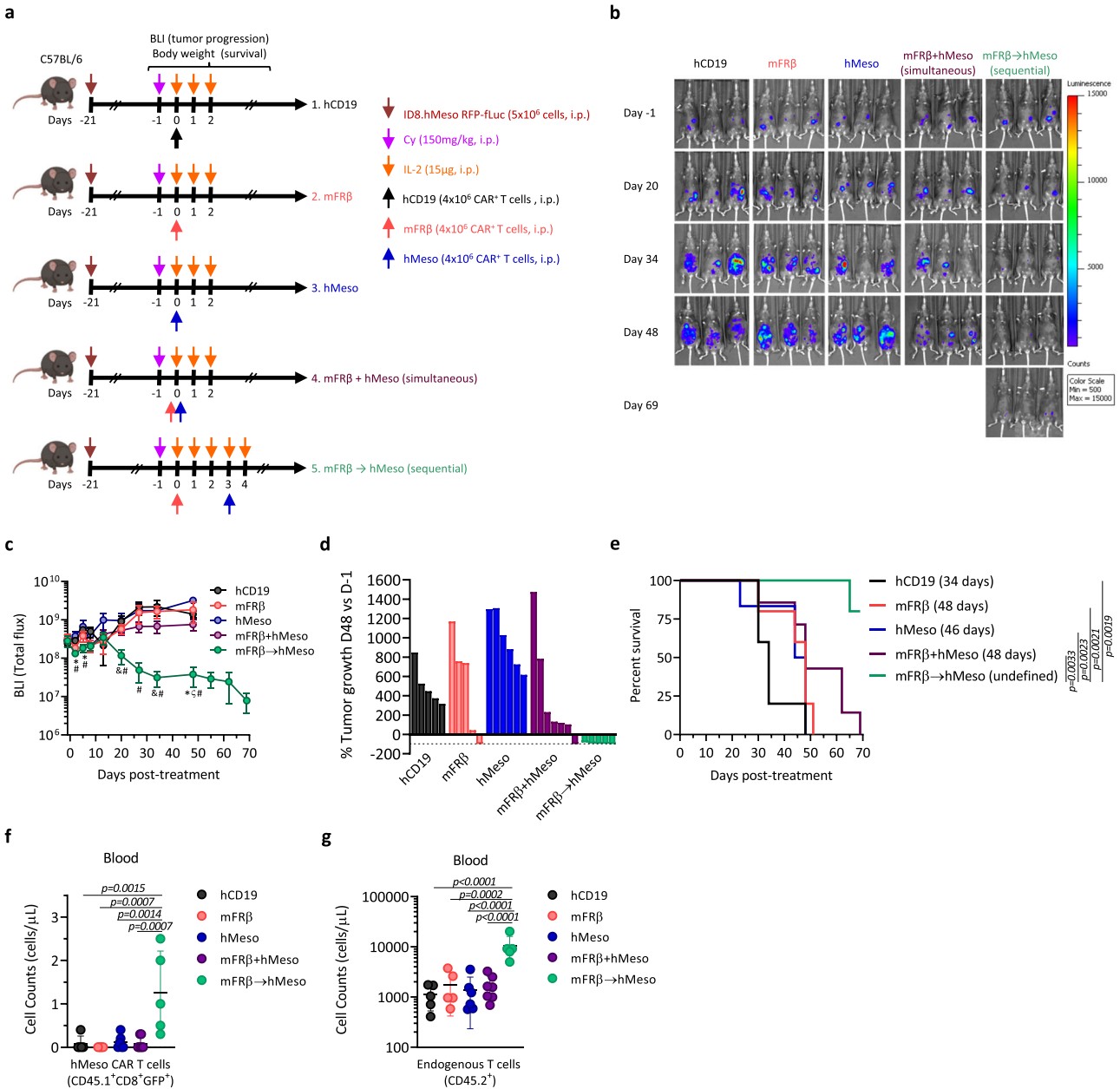

**Fig. 6 Preconditioning the TME with anti-TAM CAR-T cells improves the efficacy of tumor-specific CAR-T cells. a** Schematic representation of the in vivo study. Groups treated with only one of the CARs received an equivalent dose of UTD T cells to equal the total dose. hMeso CAR-T cells were prepared from CD45.1 donor mice (n = 5 mice for hCD19, mFRβ, and mFRβ → hMeso groups, 6 for hMeso, and 7 for mFRβ + hMeso). **b** BLI images from representative mice at indicated time points are shown. **c** Quantification of BLI shown as mean radiance ± SEM. **d** Waterfall plot showing the percentage of change in tumor BLI on day 48 vs. baseline (day −1). Data from individual mice are represented. **e** Kaplan–Meier survival curves are represented. The endpoint was defined at animal body weight increase >10% due to ascites formation. Mean survival is indicated in the legend. The concentration of **f** hMeso CAR+ T cells (live, CD45.1+/CD8+/GFP+) and **g** endogenous T cells (live, CD45.2+) were quantified in the blood of mice 9 days after administration of hMeso CAR-T cells and represented as mean ± SD. Experiment represented in panels (**a–e**) was performed twice. **c** P values were calculated by a multiple two-tailed t test (mFRβ → hMeso as compared to (*) hCD19; (&) mFRβ; (ζ) hMeso; or (#) mFRβ + hMeso) and exact values are indicated as follow: D2 * = 0.007, =0.016; D5 * = 0.015, # = 0.036; D20 & = 0.046, # = 0.021; D27 # = 0.019; D34 & = 0.047, # = 0.040; D48 * = 0.020, ζ = 0.004, # = 0.024. P values by a **e** log-rank Mantel–Cox test or **f, g** a one-way ANOVA with Tukey's multiple comparison test are indicated. Source data are provided in the Source Data file.

cancer. The accumulation of FRβ+ TAMs appears to be influenced at least, in part, by the presence of soluble factors produced by tumor cells in the TME that favor the polarization and recruitment of M2-like macrophages. In our studies, FRβ expression was significantly upregulated in vitro by culturing peritoneal macrophages from naive mice in the presence of supernatants from ID8 tumor cell cultures. This observation is similar to the results obtained by Puig-Kröger et al.[25] using in vitro-polarized human macrophages and in alignment with findings from evaluations of nearly a thousand tumor sections, including 20 different human cancer types, in which FRβ expression on stromal cells (primarily TAMs), correlated positively with cancer stage and the presence of lymph node metastasis[34]. In addition, our findings showing a correlation between

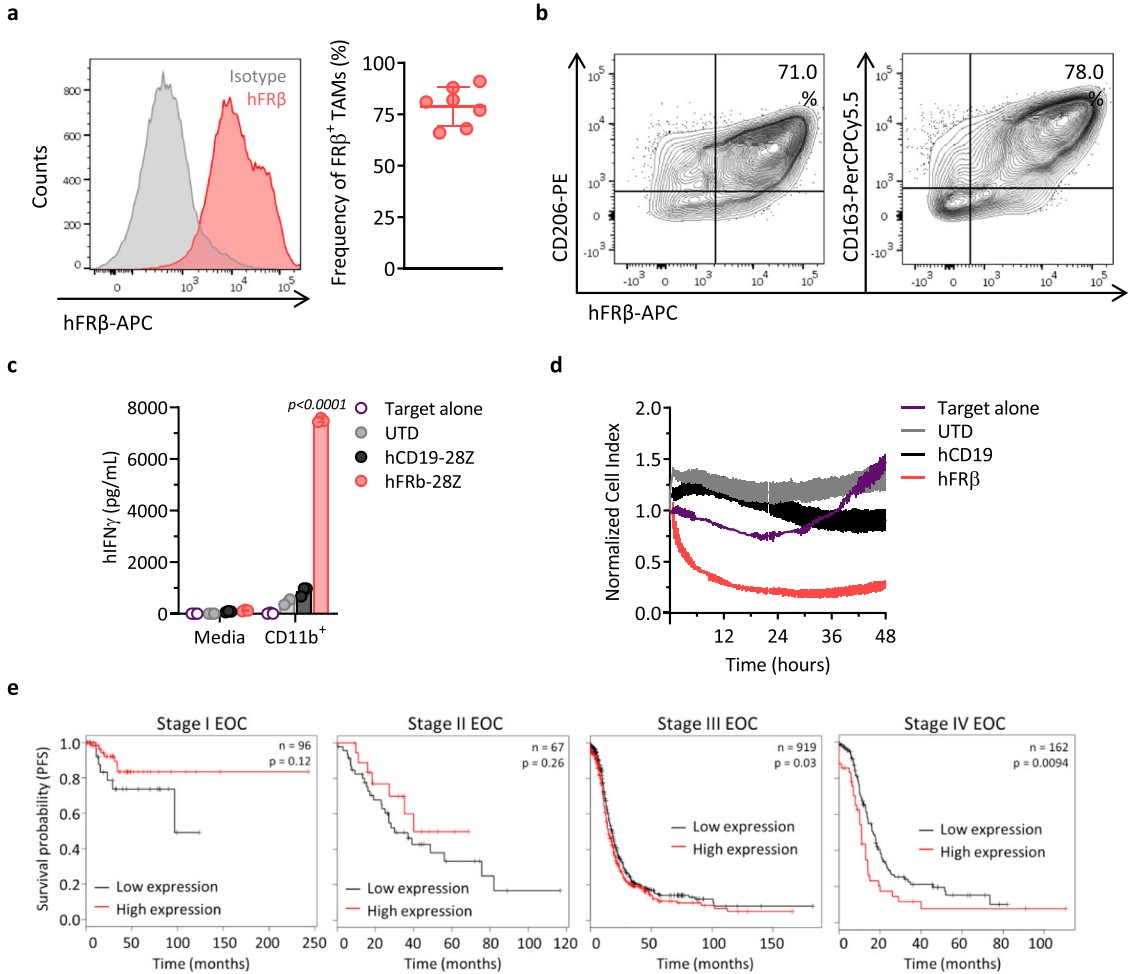

**Fig. 7 Ovarian cancer TAMs highly express FRβ and are targeted by human FRβ CAR-T cells. a** hFRβ expression in TAMs obtained from the ascites of ovarian cancer patients as assessed by flow cytometry. Representative histogram (gray—isotype, red—anti-hFRβ, left panel) and mean frequency of FRβ⁺ TAMs (defined as live, CD45⁺CD11b⁺CD14⁺ cells, right panel) ± SD are shown (n = 7 ascites biospecimens). **b** Representative flow plots showing coexpression of hFRβ (*X*-axis) and the surface markers CD206 (left panel) and CD163 (right panel, *Y*-axis) in ovarian cancer patient TAMs. Overnight cocultures of hFRβ CAR, hCD19 CAR, or UTD T cells with CD11b⁺ cells isolated from the ascites of ovarian cancer patients were established at a 1:1 E:T ratio. **c** Levels of secreted hIFN-γ as detected by ELISA in the supernatants. *P* values by a two-way ANOVA with Tukey's multiple comparison test are indicated. **d** Real-time cytotoxicity assay performed using xCelligence technology. **c, d** Representative data from one of three different patients is shown as mean ± SD of triplicates. **e** Progression-free survival curves for human ovarian cancer patients with high (red) or low (black) expression of *FOLR2* (gene encoding for FRβ) at different clinical stages. The clinical outcome data were retrieved from the Kaplan–Meier plotter database. *N* indicates the number of patients included in the analysis for each stage. *P* values by a log-rank Mantel–Cox test are indicated. EOC epithelial ovarian cancer. Source data are provided in the Source Data file.

high *FOLR2* gene expression and worse PFS only in later stages of ovarian cancer further supports this notion. Together, these results suggest a dynamic reprogramming of TAM phenotype during tumor progression, an observation also made by others evaluating various M2 markers in human TAMs[60–62].

Striking differences existed between fresh FRβ⁻ and FRβ⁺ populations of TAMs isolated from tumor-bearing mice. FRβ⁺ TAMs displayed a morphology and metabolic profile that was characteristic of an M2 phenotype, while FRβ⁻ TAMs exhibited characteristics consistent with M1 macrophages[35,49]. Differential transcriptomic profiles accompanied these morphologic and metabolic differences, with upregulated expression of an array of M2-like genes and concomitant downregulation of M1-like genes in the FRβ⁺ TAM population, as compared to their FRβ⁻ counterparts. The secretory profile of both populations supported this M1/M2 classification, with FRβ⁻ TAMs preferentially secreting proinflammatory cytokines and FRβ⁺ TAMs secreting the immunosuppressive cytokine IL-10. Moreover, FRβ⁺ but not FRβ⁻ TAMs inhibited the proliferation and cytokine secretion by tumor-specific T cells. Our results, therefore, suggest that the accumulation of FRβ-expressing TAMs in the TME may limit endogenous tumor-specific T cell activity, as well as activity from adoptively transferred T cells, and rationalize the development of effective FRβ⁺ TAM-depleting approaches for cancer therapy.

Here, we bring forward a technology based on the use of CAR-T cell therapy to specifically target and deplete protumorigenic TAMs. Our data show that FRβ⁺ TAMs can be selectively and efficiently depleted in vivo through the administration of FRβ-specific CAR-T cells, resulting in controlled tumor progression in three different preclinical tumor models and improved survival in the ID8 ovarian cancer model. The antitumor efficacy of this CAR approach was achieved by the depletion of FRβ⁺ TAMs, and not by the direct targeting of tumor cells themselves, which did not express the targeted antigen. Rather, specific depletion of the immunosuppressive TAM subset by FRβ CAR-T cells re-conditioned the TME, fostering a terrain enriched with proinflammatory myeloid cells and

enhancing endogenous antitumor T cell immunity. This is supported by the observation that FRβ[+] TAM depletion promoted the recruitment of inflammatory monocytes and endogenous tumor-specific cytotoxic CD8[+] T cells to the TME, consistent with previous studies using alternative strategies to deplete TAMs[63,64]. Endogenous CD8[+] T cells were imperative for the antitumor activity exerted by FRβ CAR-T cells, as this effect was abrogated in mice lacking functional CD8[+] T cells.

The use of CAR-T cells to deplete macrophages is an alternative approach to previously reported strategies to target TAMs with monoclonal antibodies or small molecules that rely on continuous administration of the agents to prevent rapid macrophage rebound[65]. In the clinical setting, we might predict a prolonged effect mediated by CAR-T cells, as they are living drugs that have the capacity to proliferate and to persist after their infusion[66–68], allowing for long-term, antigen-specific targeting[69]. The impact of sustained depletion of FRβ-expressing TAMs anticipated by using the human version of the FRβ-specific CAR is currently unknown, and the relatively short persistence of mouse CAR-T cells[18,70,71] has limited the evaluation of possible long-term effects in syngeneic mouse models. Another concern might be the expression of FRβ in activated macrophages in several autoimmune diseases such as RA[72–75]. The use of agents that reduce this population of activated macrophages ameliorated symptoms in animal models of inflammatory disease, demonstrating the therapeutic benefit of targeting FRβ[+] macrophages also in the setting of inflammatory diseases without deregulating autoimmunity[76–81]. In any case, the use of messenger RNA (mRNA) electroporation for transient expression of the CAR or the incorporation of suicide genes to eliminate T cells stably expressing the CAR in the event of CAR-related toxicities provide options for mitigating risk [23].

Reproducible and significant delay of tumor progression was induced by mFRβ CAR-T cell monotherapy; however, complete cures were not observed, suggesting the need for combinatorial approaches. CAR-T cell therapy for solid tumors has yet to recapitulate the unprecedented clinical responses achieved in hematologic malignancies[82–85], and complete responses to CAR-T cell therapy have been anecdotal in solid tumor patients[86–88]. Accordingly, the idea of altering the TME by using engineered T cells in order to improve the efficacy of conventional tumor-targeted CAR-T cells has become increasingly attractive[16,89]. One study reports a CD123-specific CAR that could simultaneously target tumor cells and TAMs in Hodgkin lymphoma[90]. In that study, the use of immunodeficient mouse models limited the ability to explore the effect of multiple endogenous immune components and the role of the acquired immune system, rationalizing the use of syngeneic models as applied in our study. In immunocompetent mice, we found that selective depletion of FRβ[+] TAMs prior to the administration of mesothelin-specific CAR-T cells significantly improved engraftment and expansion of tumor-specific CAR-T cells and promoted a more robust endogenous T cell recruitment. This resulted in complete tumor regressions and prolonged survival, while simultaneous administration of both CARs did not provide a significant benefit. Although the precise mechanisms responsible for the requirement of sequential treatment cannot be defined due to limitations of the model system, the findings derived from FRβ T cell monotherapy studies suggest that time is required for the immunosuppressive TME to shift towards one that supports productive antitumor immune responses. In a series of pilot clinical trials of tumor-infiltrating lymphocyte therapy, the reported objective responses rates increased when a preconditioning regimen was introduced and intensified[91–93]. This is aligned with our observation of improved efficacy of mesothelin-specific CAR therapy when TAM depletion following Cy chemotherapy is provided as a preparative treatment regimen. We, therefore, postulate that the

development of a staged clinical regimen that first pre-conditions the TME by a combination of lympho-depleting chemotherapy and specific depletion of immunosuppressive FRβ[+] TAMs and is followed by administration of a conventional tumor-targeted CAR-T cell product may enhance outcomes in patients with most solid tumors. These findings may also be generalizable to the use of alternative targets[94] or macrophage disrupting agents combined with other forms of T cell-provoking immunotherapies.

Supporting the translatability of this CAR-T cell-based approach to humans, our results demonstrated the presence of M2-like FRβ[+] TAMs in ascite samples from ovarian cancer patients, as well as the feasibility of their targeting with hFRβ CAR-T cells. In addition, our results in melanoma and colon adenocarcinoma mouse models together with patient survival data suggest that this treatment modality would not be restricted solely to ovarian cancer patients, but rather depletion of FRβ[+] TAMs may have clinical impact in patients suffering from other types of solid tumor.

## Methods

**Cell lines.** All cells were grown in complete media (CM, RPMI-1640-GlutaMAX with 100 μg/mL streptomycin, 10 μg/mL penicillin, and 10% fetal bovine serum), unless otherwise noted. The retroviral packaging cell line PlatE with ecotropic envelope and the murine ovarian carcinoma cell lines ID8 and ID8.OVA were kindly provided by George Coukos[95]. The parental ID8 line was stably transduced with a lentiviral construct containing mCherry (RFP) and fLuc separated by a viral T2A ribosomal skipping element (RFP-2A-fLuc) to create ID8 RFP-fLuc. ID8 RFP-fLuc cell line was then stably transduced with lentiviral constructs encoding murine FRβ complementary DNA (Origene) or hMeso complementary DNA (kindly provided by Steven Albelda) to produce ID8.mFRβ RFP-fLuc and ID8.hMeso RFP-fLuc, respectively. MC38 colon adenocarcinoma cell line was purchased from Kerafast. B16-F10 melanoma cell line was purchased from ATCC.

**TAM isolation.** Six- to eight-week-old female C57BL/6 mice were purchased from Charles River, housed and treated under the University of Pennsylvania Institutional Animal Care and Use Committee-approved protocols. SWV FRβ KO or WT mice were kindly provided by Richard H. Finnel (Baylor College of Medicine, Houston) and were breeded in-house[96]. Six- to eight-week-old male and female SWV were used.

Mice were inoculated i.p. with $5 \times 10^6$ ID8 RFP-fLuc and euthanized at indicated time points to collect tumor ascites by peritoneal wash. Ten milliliters of phosphate-buffered saline (PBS) were injected i.p. and total cells in the wash were collected. Red blood cells were lysed using ACK (ammonium-chloride-potassium) lysis buffer (Thermo Fisher). For flow cytometry assays, $1–2 \times 10^6$ total cells were stained and analyzed as described below. For in vitro coculture assays, cells were labeled with CD11b MicroBeads (mouse/human) (Miltenyi Biotec) and isolated with LS MACS separation columns according to the manufacturer's instructions. CD11b[+] cells were stained for markers CD45 and F4/80, and live, double-positive cells were sorted based on FRβ expression by the Flow Cytometry and Cell Sorting Facility (UPENN).

**Upregulation of FRβ in vitro.** Peritoneal cells were collected from naive mice by peritoneal lavage with PBS, split into two groups, and plated in 6-well plates. Once the macrophages were adhered, cells were washed once with PBS and incubated at 37 °C in the presence of CM, or 72 h ID8 tumor cell culture supernatant. After 72 h, cells were detached by using Accutase (Sigma-Aldrich) and stained for flow cytometry as described below.

**Flow cytometry analysis.** Up to $2 \times 10^6$ cells were labeled per tube in staining buffer (2% fetal bovine serum/PBS). Fc receptors were blocked by using TruStain FcX PLUS (anti-mouse CD16/32, #156603, 1:200 dilution) antibody as indicated by the manufacturer (BioLegend). The following antibodies were used in this study: hFRβ-APC (#391705, clone 94b, dilution 1:50), hCD206-PE (#321105, clone 15-2, dilution 1:50), human CD163-PerCPCy5.5 (#333607, clone GHI/61, dilution 1:50), mouse FRβ purified (#153302, clone 10/FR2, dilution 1:400), mouse FRβ-APC (#153305, clone 10/FR2, dilution 1:40), mouse CD45-FITC (#103107, clone 30-F11, dilution 1:200), mouse CD45-PE (#103105, clone 30-F11, dilution 1:80), mouse CD45-PacBlue (#103125, clone 30-F11, dilution 1:200), mouse CD45-APCCy7 (#103115, clone 30-F11, dilution 1:80), mouse CD45.1-Pacific Blue (#110722, clone A20, dilution 1:50), mouse CD45.2-BV711 (#109847, clone 104, dilution 1:80), mouse CD3-PECy7 (#100219, clone 17A2, dilution 1:80), mouse CD8α-APC (#100711, clone 53-6.7, dilution 1:80), mouse CD8α-APCCy7 (#100713, clone 53-6.7, dilution 1:20), mouse CD4-PE (#116005, clone RM4-4, dilution 1:80), mouse/human CD11b⁻PacBlue (#101223, clone M1/70, dilution 1:50), mouse/human CD11b⁻BV650 (#101237, clone M1/70, dilution 1:80), mouse F4/80-PECy7 (#123113, clone BM8, dilution 1:80), mouse CD206-AlexaF700 (#141733, clone

C068C2, dilution 1:200), mouse PD-L1-PercCPCy5.5 (#124333, clone 10F.9G2, dilution 1:20), mouse PD-1-BV605 (#135219, clone 29F.1A12, dilution 1:160), mouse PD-L2-PE (#107205, clone TY25, dilution 1:20), mouse Ly6C-APCCy7 (#128025, clone HK1.4, dilution 1:80), mouse Ly6G-PerCPCy5.5 (#127615, clone 1A8, dilution 1:80), mouse IFN-γ-PerCPCy5.5 (#505821, clone XMG1.2, dilution 1:20), mouse TNF-α-BV650 (#506333, clone MP6-XT22, dilution 1:40), goat anti-rat IgG-APC (#405407, clone Poly4054, dilution 1:80), mouse/rat/human FOXP3-AlexaF647 (#320014, clone 150D, dilution 1:20) (BioLegend), mouse CD204-PE (#130-120-811, clone REA148, dilution 1:50, Miltenyi), mouse Arg1-FITC (#IC5868F, dilution 1:10, R&D Systems), mouse Egr2-PE (#12-6691-82, clone erongr2, dilution 1:80, eBiosciences), iTAg Tetramer/PE-H2Kb OVA (SIINFEKL, #TB-5001-1, dilution 1:10, MBL), hMeso purified (#sc-33672, clone K1, dilution 1:100, SCBT), and mouse mesothelin purified (#LS-C179484-100, clone B35, dilution 1:100, LifeSpan Biosciences). The respective isotypes were also purchased. LIVE/DEAD Fixable Aqua Dead Cell Stain Kit (#L34957, dilution 1:1000, Thermo Fisher Scientific) was used to assess viable cells. Flow cytometry was performed with a BD LSRFortessa and flow data were analyzed with FlowJo v10 6.2 (FlowJo LLC).

**Immunofluorescence.** FRβ$^+$ or FRβ$^-$ TAMs growing on glass coverslides at low confluence were fixed with 4% formaldehyde, stained with phalloidin-rhodamine (catalog # R415, Thermo Fisher Scientific), and cell nuclei were counterstained with DAPI. Slides were visualized by confocal microscopy, using a Zeiss LSM 710 microscope.

**RNA extraction, transcriptome sequencing, and RNA-sequencing (RNA-seq) analysis.** Total RNA from FACS-sorted FRβ$^+$ or FRβ$^-$ TAMs was obtained by using an RNeasy Plus Mini Kit (Qiagen) according to the manufacturer's instructions.

*Library preparation and sequencing.* Total RNA was assayed for quantity and quality with an Agilent 2100 Bioanalyzer instrument using the RNA 6000 Nano Kit (Agilent Technologies, Part number 5067-1512). Average RIN values were 9.786 ± 0.3. Libraries were prepared using TruSeq Stranded mRNA HT Sample Prep Kit (Illumina, Part number 20020595) as per standard protocol in the kit's sample prep guide. Libraries were assayed for overall quality using HS D5000 ScreenTape assay (Agilent Technologies, Part numbers 5067-5592 and 5067-5593) of Agilent 2200 TapeStation System. Average library molarity was 13.9 ± 4.2 nM. Samples were multiplexed for sequencing. A MiSeq micro test lane was run to check pool balance followed by 100 bp single-read sequencing on an Illumina HiSeq 4000 sequencer. Illumina's bcl2fastq version v2.17.1.14 software was used to convert bcl to fastq files.

*RNA-seq analysis.* Reads were filtered to retain only high-quality reads. In addition, ribosomal reads and repeats were eliminated by alignment to a generic set of ribosomal/repeat sequences. Remaining reads were processed with RNA-Seq Unified Mapper (RUM)[97]. Following RUM alignments to mouse genome (build mm10) and known transcripts (RefSeq, UCSC, ENSEMBL, etc.), a feature-level quantitation (transcript, exon, and intron) was output as a result. To analyze global gene expression profiles, the number of uniquely aligning reads to transcripts in RefSeq were extracted from the RUM output. Multidimensional scaling plots, principal coordinate analysis plots and K-means heatmaps were generated using quantile normalization of log 2 counts. Differential expression analysis was performed (R library edgeR[98]) using a design formula that included the mouse identifier as well as the *Folr2* (FRβ) status of the cells. Standard processing steps using dispersion estimation, generalized linear model fit, and log-ratio testing were performed. The resulting p values were adjusted for multiple testing using the default method (Benjamini and Hochberg) for the topTags function. R statistics programming was used to build log 2(fold change, FC) vs. log 2(counts per million) plots. Adjusted p value <0.05, absolute log 2(FC) > 0.26 and false discovery rate were considered. Heatmap of normalized mean-centered mRNA expression of selected genes was plotted using iDEP.90[99].

**Analysis of metabolic parameters.** Mitochondrial function was assessed with an extracellular flux analyzer (Seahorse Bioscience). Individual wells of XF96 cell culture microplates were coated with CellTak in accordance with the manufacturer's instructions. The matrix was adsorbed overnight at 37 °C, aspirated, air-dried, and stored at 4 °C until use. Macrophages were plated at 100,000 cells/well in the presence or absence of 5 ng/mL LPS (Sigma-Aldrich) or 20 ng/mL mIL-4 (PeproTech) and incubated overnight at 37 °C in CM. During instrument calibration, the cells were washed once with PBS and the medium was switched to an XF assay medium (nonbuffered RPMI-1640) containing 10 mM glucose and 2 mM ʟ-glutamine. The cells were switched to a $CO_2$-free (37 °C) incubator for 30 min. XF96 assay cartridges were calibrated in accordance with the manufacturer's instructions. Cellular OCRs and ECARs were measured under basal conditions and following treatment with 1.3 μM oligomycin, 1.5 μM FCCP (phenylhydrazone), and 500 nM rotenone/antimycin A (XF Cell Mito Stress Kit, Seahorse Bioscience). Mitochondrial mass was measured by flow cytometry after staining 100,000 cells with MitoTracker Green FM (Invitrogen) for 15 min at 37 °C and 100 nM. Mitochondrial membrane potential was measured by flow cytometry after staining

100,000 cells with tetramethylrhodamine methyl ester (Invitrogen) for 30 min at 37 °C and 25 nM.

**CAR construction and retroviral production.** MSGV1 retroviral plasmid DNA encoding the DC101 CAR with murine CD8α hinge and CD28 transmembrane followed by murine CD28, 41BB, and CD3ζ intracellular signaling domains was provided by Steve Rosenberg[20]. eGFP-T2A sequence and P4 scFv (specific for hMeso[100]) was added to create MSGV-GFP-2A-P4-m28BBZ. Plasmid DNA containing the CL10 scFv, specific for mouse FRβ, was kindly provided by Takami Matsuyama[29]. The CL10 scFv was amplified using two-step PCR first with primers CL10Fwd: 5′-agcaactgcaactggagtacattcagacattgtgatgacccaatctccatcctctctgg-3′ and CL10Rev: 5′-tatgcggccgctgaggagacagtgactgaagctccttgaccccaggcatccataatatcg-3′, and second with primers LeadFwd: 5′-tatccatgggatggtcatgtatcatcctttttctggtagcaactg-caactggagtacattc-3′ and CL10Rev to add on sequence encoding the DC101 leader sequence from the DC101 vector and restriction sites for *Nco*I and *Not*I. The purified PCR product and MSGV-GFP-2A-P4-m28BBZ plasmid DNA were digested with the relevant enzymes (NEB), gel purified, and ligated at a 3:1 insert: vector ratio using the Rapid DNA Ligation Kit (Roche) to create MSGV-GFP-2A-CL10-m28BBZ. MSGV-GFP-2A-CD19-m28BBZ was created using an analogous strategy and primers hCD19Fwd: 5′-agcaactgcaactggagtacattcagacatccacag cagac tacatcctccctgtctg-3′, hCD19Rev: 5′-tatgcggccgctgaggagacggtgactgaggttccttggccccagta gtccatagcatag-3′ and LeadFwd to amplify the FMC63 scFv specific for human CD19[101]. MSGV-GFP-2A-M11-m28BBZ was created using primers M11Fwd: 5′-agcaactgcaactggagtacattcacaagtccaattgcagcagagcggagcagaagtg-3′, M11Rev: 5′- tatgcggccgctttgatttcaccttggttccaggaccgaaatccggggtggtgtacgtc-3′ and LeadFwd to amplify the M11 scFv specific for hMeso[102]. MSGV-GFP-2A-A03-m28BBZ was created using primers A03Fwd: 5′-agcaactgcaactggagtacattcaatggccagcccctgaccagatt cctgagcctg-3′, A03Rev: 5′-tatgcggccgctctggggggccagggtctgcatgtgcagggggtcgtaggtgtcct tggtg-3′ and LeadFwd to amplify the A03 scFv specific for mouse mesothelin[103]. DNA sequencing was used to confirm the expected sequence. Hi-Speed MIDI or MAXI DNA Prep Kits (Qiagen) were used to produce high-quality DNA. A total of $6 \times 10^6$ PlatE cells in log-phase growth were plated in T150 tissue culture flasks in 27 mL CM. After 24 h, each flask was transduced with 20 μg MSGV1 retroviral transfer plasmid DNA using 3 mL Opti-MEM (Gibco) and Lipofectamine 2000 (Invitrogen) at a 3:1 Lipofectamine:DNA ratio. Thirty milliliters of CM was replaced at 24 and 48 h. Supernatants from 48 and 72 h post transfection were collected, 0.45 μm filtered, and frozen at −80 °C until use. CARs targeting human CD19 (FMC63 scFv) or hFRβ (m923 scFv) including CD28 and CD3ζ human intracellular signaling domains were previously reported [23,101].

**Generation of mouse CAR-T cells.** Mouse T cells were cultured in mouse T cell media (CM with 50 μM β-mercaptoethanol, 100 mM sodium pyruvate, 1× Gluta-MAX, and 50 IU/mL murine IL-2 (Peprotech)). Spleens were isolated from C57BL/ 6 or B6 Cd45.1 mice as indicated after humane euthanasia according to protocols approved by the University of Pennsylvania Institutional Animal Care and Use Committee. Splenocytes were dissociated and pushed through a 70-μm cell strainer. Red blood cells were lysed using ACK lysis buffer and cell number was determined. A total of $3 \times 10^6$ total splenocytes/mL/well were activated with anti-mouse CD3/CD28 antibody-coated beads (Dynabeads, Invitrogen) at a 1.33:1 bead: cell ratio in 24-well plates. Splenocytes were transduced with retroviral vectors on days 1 and 2 post activation. For retroviral transduction, 0.5 mL RetroNectin (Takara) diluted to 25 μg/mL in sterile PBS was immobilized overnight at 4 °C in 24-well nontissue culture-treated plates. After overnight incubation, wells were washed with PBS and blocked for 10 min with 2% bovine serum albumin/PBS. Three milliliters of 48 h retroviral supernatant was added per well and plates were centrifuged at $2000 \times g$ for 1.5 h at room temperature. The supernatant was removed and 0.5 mL $(1.5 \times 10^6)$ day 1 activated splenocytes were added per well. Cells were centrifuged for an additional 10 min at $1000 \times g$ and returned to 37 °C. Transduction was repeated on day 2 post activation using 72 h retroviral supernatant. For UTD T cells, CM was used in place of retroviral supernatant. T cell media were added daily to maintain ~$1 \times 10^6$ cells/mL. Beads were removed on day 4. T cells were used for in vitro or in vivo assays on days 5–7 as indicated. The presence of surface CAR expression was measured in transduced T cells on day 5 post activation by flow cytometry. mFRβ CAR expression was measured using biotinylated rabbit anti-human IgG (H + L) and hCD19 CAR was measured using biotinylated goat anti-mouse IgG (H + L) (Jackson Immunoresearch). Cells were washed and secondary labeled with streptavidin-APC. Coexpression of GFP and APC was used to determine surface CAR expression in transduced T cells. Binding of mFRβ CAR to biotinylated recombinant protein antigen was also evaluated. Recombinant murine FRβ was purchased from R&D. Recombinant hMeso was used as unspecific control and was produced in yeast. Recombinant proteins were biotinylated in-house using EZ-link-Sulfo-NHS-LC-biotin (Thermo) and purified with dialysis using Tube-O-Dialyzer Medi (G-Biosciences). T cells were labeled with 500 ng biotinylated recombinant protein for 30 min at 4 °C. Cells were washed and secondary labeling with streptavidin-APC was conducted for 25 min at 4 °C.

**Immunosuppression assays.** A total of 50,000 FACS-sorted FRβ$^+$ or FRβ$^-$ TAMs were cocultured with 50,000 ID8.hMeso cells in 96-well plates. After overnight incubation at 37 °C, 50,000 hMeso-specific CAR-T cells (P4 scFv), which were

previously labeled with CellTrace Violet (Thermo Fisher) according to the manufacturer's instructions, were added to the cocultures. Where indicated, mIL-10R blocking antibody or the appropriate isotype control (BioLegend) were added to the cocultures at 25 μg/mL. At 72 h, T cells were harvested and analyzed by flow cytometry. Supernatants from the same cocultures were collected and assayed for mIFN-γ or mIL-10 presence by using ELISA (enzyme-linked immunosorbent assay) Kits (BioLegend) according to the manufacturer's instructions.

A total of 20,000 FACS-sorted FRβ$^+$ or FRβ$^-$ macrophages were cocultured with 100,000 carboxyfluorescein succinimidyl ester-labeled OT-1 splenocytes in the presence of specific OVA$_{257-264}$ or unspecific MART1 peptides (GenScript) at 1 μM. As positive control to stimulate T cells, cell activation cocktail (R&D Systems) was used. At 72 h, T cells were harvested, stained for viability and T cell marker CD3, and analyzed by flow cytometry. Supernatants from the same cocultures were collected and assayed for mIFN-γ presence by using ELISA Kit (BioLegend) according to the manufacturer's instructions. Alternatively, the number of mIFN-γ-producing cells was measured using an ELISpot assay. Briefly, cocultures were established as described above in coated ELISPOT plates. After overnight incubation, ELISpot assay was performed using a mIFN-γ ELISpot set according to the manufacturer's instructions (BD Biosciences). Spots were then counted using an automated ELISpot reader (AutoimmunDiagnostika GmbH).

**Cytokine release assays.** For assessment of mIFN-γ production in response to an immobilized protein antigen, 1:2 dilutions of recombinant murine FRβ (R&D Systems) in 100 μL PBS were plated in 96-well ELISA plates in triplicate wells per condition (range 31–1000 ng/well). After overnight coating at 4 °C, wells were washed with PBS and 100,000 CAR$^+$ (or UTD) mouse T cells were added in 200 μL CM. After 18 h culture, cell-free supernatant was harvested. For assessment of mIFN-γ production in response to target tumor cells, ID8 or ID8.mFRβ cells were plated at 100,000 cells/well in 96-well plates, alone or in the presence of 100,000 CAR$^+$ (or UTD) mouse T cells (E:T ratio 1:1), and incubated at 37 °C. CD11b$^+$ cells isolated from the ascites of FRβ KO or WT SWV mice bearing ID8 ascites were plated at 50,000 cells/well in 96-well plates, alone or in the presence of 50,000 CAR$^+$ (or UTD) mouse T cells (E:T ratio 1:1), and incubated at 37 °C. After 24 h, supernatants were collected and assayed for the presence of mIL-10 and/or mIFN-γ by using ELISA Kits following the manufacturer's recommendations (BioLegend). Supernatants from flow-sorted TAMs plated at 50,000 cells/well and cultured overnight in the presence or absence of 5 ng/mL LPS (Sigma-Aldrich) were also assayed for the presence of multiple cytokines by using a mouse inflammation panel LEGENDplex bead-based assay (BioLegend). For experiments with human ascites samples, CD11b$^+$ or CD11b$^-$ cells were plated at 100,000 cells/well in 96-well plates, alone or in the presence of 100,000 CAR$^+$ (or UTD) human T cells (E:T ratio 1:1), and incubated at 37 °C. After 24 h, supernatants were collected and assayed for the presence of hIFN-γ by using ELISA Kits following the manufacturer's recommendations (BioLegend).

**Cytotoxicity assays.** Cytotoxic killing of target cells was assessed using the xCELLigence Real-Time Cell Analyzer System (ACEA Biosciences). Tumor cells ID8, ID8.mFRβ, or ID8-hMeso were plated at $1 \times 10^4$ cells/well. CD11b$^+$ cells isolated from the ascites of FRβ KO or WT SWV mice bearing ID8 ascites were plated at 50,000 cells/well. For experiments with human ascites, isolated CD11b$^+$ cells were plated at 100,000 cells/well. After overnight cell adherence, CAR$^+$ (or UTD) T cells were added at a 1:1 E:T ratio (or 10:1 E:T ratio, as indicated in the figure legends). Cell index (relative cell impedance) was monitored every 20 min and normalized to the maximum cell index value immediately prior to effector-cell plating. The percentage of cytolysis was calculated at indicated time points using the RTCA software 2.0.

**In vivo tumor models.** All animal experiments were done according to protocols approved by the Institutional Animal Care and Use Committee at the University of Pennsylvania. Animal housing was conducted in accordance with recommendations in the Guide for the Care and Use of Laboratory Animals of the National Institutes of Health, with a 12 h light/dark cycle and a temperature range of 68–74 °F. Six- to eight-week-old female C57BL/6 mice were purchased from Charles River. B6.129S2-Cd8$^{tm1Mak}$ (CD8 KO) and B6.SJL-Ptprc$^a$ Pepc$^b$/BoyJ (CD45.1) mice were purchased from The Jackson Laboratory. Six- to seven-week-old male and female SWV FRβ KO or WT mice were breed in-house.

For the assessment of antitumor efficacy of mFRβ CAR$^+$ T cells on a mFRβ-expressing tumor model, C57BL/6 mice were implanted i.p. with $5 \times 10^6$ ID8.mFRβ RFP$^-$fLuc tumor cells. On days 4, 10, 17, 31, and 38 following tumor injection, mice received $5 \times 10^6$ hMeso CAR-T cells (as control) or mFRβ CAR$^+$ T cells. For the rest of experiments, mice were inoculated i.p. with $5 \times 10^6$ ID8 RFP-fLuc, ID8. hMeso RFP-fLuc or ID8-OVA cells. A total of $4 \times 10^6$ or $8 \times 10^6$ CAR$^+$ T cells (as indicated in the figure legends) were injected in 200 μL of PBS 21 days after tumor inoculation. Cy (150 mg/kg) was injected i.p. one day before T cell transfer and hIL-2 (Proleukin) was provided at 15 μg/dose in 100 μL i.p. for 3 consecutive days following T cell transfer. Tumor growth was monitored by bioluminescent imaging and weight gain as a surrogate marker for ascites formation. Mice were euthanized at indicated time points for analysis or when they had gained >20% of initial body weight. Bioluminescence imaging was performed by using IVIS

Spectrum Imaging System and quantified with the Living Image Software (PerkinElmer). Mice were given an i.p. injection of 150 mg/kg D-luciferin (Caliper Life Sciences, Hopkinton, MA) and imaged under isoflurane anesthesia at the peak of photon emission. Survival curves were graphed considering an intermediary endpoint of weight gain >10% of initial body weight. Peripheral blood was collected via retro-orbital blood collection under isoflurane anesthesia. Fifty microliters per sample were labeled for CD45, CD11b, Ly6G, Ly6C, CD3, CD4 and CD8 and cell counts/μL of blood were calculated using Trucount tubes (BD Biosciences).

For studies with subcutaneous solid tumor models, C57BL/6 mice were inoculated subcutaneously with $0.25 \times 10^6$ B16 melanoma or $0.5 \times 10^6$ MC38 colon adenocarcinoma cells. After 14 days, tumors were collected and mechanically dissociated using gentleMACs tubes and dissociator according to the manufacturer's recommendations (Miltenyi Biotec). Single-cell suspensions were stained for FACS analysis as described. For antitumor efficacy experiments, 10 days following tumor inoculation, mice were randomized in groups and treated with a single intravenous dose of $8 \times 10^6$ mouse CAR$^+$ T cells, accompanied with Cy + IL-2 conditioning as indicated above. Tumors were measured twice a week with caliper, and volumes were calculated as $V = 1/2 \times \text{length} (L) \times \text{width} (W) \times W$. Mice were euthanized when tumor diameter was ≥2 cm.

**Primary ovarian cancer patient samples.** Randomly selected, de-identified ascite samples from ovarian cancer patients were purchased from the University of Pennsylvania Tumor BioTrust Collection, where ovarian cancer samples are collected under an IRB-approved research protocol (IRB702679). Written informed consent was obtained from each patient. Single-cell suspensions from liquid tumor ascites were labeled with antibodies and analyzed by flow cytometry. For in vitro assays, cells were labeled with CD11b MicroBeads (mouse/human) (Miltenyi Biotec) and isolated with LS MACS separation columns according to the manufacturer's instructions. CD11b$^+$ and CD11b$^-$ fractions were cocultured with CAR-T cells for cytokine release and/or lytic assays as described above.

**Kaplan–Meier plotter database analysis.** The Kaplan–Meier plotter (http://kmplot.com/analysis/) was used to evaluate the effect of FOLR2 gene expression on patient survival. Data from 1287 ovarian cancer patients were downloaded from Gene Expression Omnibus and The Cancer Genome Atlas (Affymetrix HG-U133A, HG-U133A 2.0, and HG-U133 Plus 2.0 microarrays)[57]. Patients were separated into two groups based on median expression of FOLR2 (probe set 204829_s_at). These groups were then compared using PFS for patients at each stage of the disease (I, II, III, and IV), and including endometroid and serous histologies. Kaplan–Meier survival plots were generated and significance by log-rank $p$ value was computed.

**Statistical analysis.** Data processing was performed with Microsoft Excel 2016. Graphing and statistical analysis were performed with GraphPad Prism 8.0 software (La Jolla, CA). Data were reported as mean ± standard deviation, unless otherwise noted. Statistical analysis was performed using one- or two-way analysis of variance with Tukey's or Sidak's correction for multiple comparisons, when appropriate, as indicated in the figure legends. When multiple time points were compared, multiple Student's $t$ test was used. Survival curves were compared using the log-rank test. A $p$ value < 0.05 means statistically significant. $P$ values are indicated in the figures and "n.s." means nonsignificant ($p > 0.05$).

**Reporting summary.** Further information on research design is available in the Nature Research Reporting Summary linked to this article.

## Data availability
The authors declare that all data supporting the results in this study are available within the paper and its Supplementary information. The RNA-seq data have been deposited in the Gene Expression Omnibus database, under accession number GSE155841. For survival analysis of ovarian cancer patients, we used compiled datasets by Kaplan–Meier plotter (KM plotter). The following datasets were used in KM plotter: GSE14764, GSE15622, GSE18520, GSE19829, GSE23554, GSE26193, GSE26712, GSE27651, GSE30161, GSE3149, GSE51373, GSE63885, GSE65986, GSE9891, and TCGA [https://www.cancer.gov/tcga]. Other data are available from the corresponding authors upon reasonable request. Source data are provided with this paper.

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

## Acknowledgements

We thank P. Halberg from the University of Pennsylvania Flow Cytometry and Cell Sorting Facility for assistance with cell sorting, and J. Schug from the Next-Generation Sequencing Core (NGSC) for assistance with RNA-seq. We also acknowledge the Small Animal Imaging Core Facility (SAIF), the Stem Cell and Xenograft Core (SCXC), the Human Immunology Core (HIC), the OCRC Tumor Bank, and the Pathology Core from the Children's Hospital of Philadelphia. We also thank G. Coukos for providing ID8. OVA cells, R. Finnel for providing SWV FRβ knockout mice, K. Byrne for providing OT-1 transgenic mice, J. Scholler and M. Siurala for providing the mesothelin CAR constructs, and Novartis Institutes for Biomedical Research. Images in Figs. 5a and 6a were created with BioRender. This work was supported by the Sandy Rollman Ovarian Cancer Foundation (SROCF) (grant number 570488; to A.R.-G.) and RO1 (grant number CA226983-02; to R.S.O'C), D.J.P. has received funding from the Ovarian Cancer Research Alliance and sponsored research funding provided by Tmunity Therapeutics, Philadelphia, PA, for this work.

## Author contributions

A.R.-G. designed and performed experiments, analyzed data, and wrote the manuscript. R.C.L. developed the mouse CAR constructs and designed and performed experiments. M.P., M.A.E., L.C.S., and N.G.M. assisted with in vivo experiments. R.S.O'C. performed metabolic experiments. V.C.-M. performed immunofluorescence and image acquisition. G.L. made RNA-seq plots. T.M. initially developed and provided mFRβ scFv. D.J.P. supervised the project, including design of experiments, data analysis, and manuscript writing.

## Competing interests

D.J.P. and the University of Pennsylvania hold published patents on the use of human FRβ-specific (US20140286973A1) and mesothelin-specific chimeric antigen receptors (WO2015090230A1). An FRβ-specific CAR (US20140286973A1) is the subject of a licensing agreement between the University of Pennsylvania and Tmunity Therapeutics. These CARs are designed for potential application in human subjects. The CL10 FRβ-specific CAR applied in this manuscript is not covered by the patent application. The other authors declare no competing interests.
