## [Peer Review File · Nature Communications]

REVIEWER COMMENTS

Reviewer #1 (Remarks to the Author):

The manuscript by Rodríguez-García et al. addresses the use of CAR T-cell mediated depletion of tumor-associated macrophages (TAM) as a feasible alternative to limit tumor growth and enhance anti-tumor immunity. The authors have used several very well-defined experimental systems (mouse tumor models) and adequately designed cellular tools, demonstrate that depletion of FOLR2+ macrophages has an impact on tumor growth, favors tumor elimination, and might complement other CAR T-cell based strategies. As a whole, the manuscript is of interest, timely, nicely introduced, and derives from previous observations of the group on the applicability of FR β -specific CAR T cells for acute myeloid leukemia. A relevant novelty of the manuscript is the identification of a marker (FOLR2) that might be useful to identify and target TAM in a wide variety of tumors.

Most of the experiments in the manuscript are clearly and properly controlled, and the manuscript includes relevant and solid supplementary information that complements the main text. As a whole, the manuscript is solid and very informative. However, a number of issues could be addressed for the manuscript to gain robustness and to fully support the authors' conclusions.

General comments:

- All throughout the manuscript, and when referring to previous reports, the authors should clearly indicate the species where the data has been generated.
- A major issue that arises from the differences at the transcriptional and cytokine levels between both subsets is the possible contribution of IL-10 to some of the observed functional effects (in fact, IL10 gene expression appears to parallel that of FOLR2). Thus, can the authors assess whether the effect of FOLR2-specific CAR-T cells is dependent on IL-10, either using IL10 KO mice or, alternatively, using anti-IL10 blocking antibodies? If so, can anti-IL-10 replace FOLR2-specific CAR-T cells in the experimental system used in Figure 6? In this regard, determination of IL10 production in the experiment presented in Figure S4 would be appropriate.
- The results from the Kaplan-Meier studies in ovarian cancer patients are particularly relevant. The authors could extend these studies to other tumor types, and determine whether this is a general effect. As an example, and given the fact that FOLR2 expression appears to be enhanced by M-CSF, a similar type of analysis could be done on breast carcinoma and using TCGA and an alternative validation cohort (Metabric?).

Specific comments:

- Figure 1 should include the presence of FOLR2+ macrophages in the peritoneum at t=0.
- The authors have gathered a very informative list of DEGs between FOLR+ and FOLR- macrophages, but the validation experiments have addressed a limited number of markers that distinguish pro- and anti-tumoral macrophages (in Figure 2). A more detailed bioinformatics analysis of this transcriptional information would enrich the manuscript, possibly also hypothesizing the factors that might be responsible for this transcriptomic differences. As an example, can the authors check whether the expression of MAF and/or MAFB is different between both macrophage subsets at the protein level? Along the same line, and given the bioenergetics data included in the manuscript, what is the level of HIF factors in both subsets? The differential expression of EglN3 suggests that this might be the case....
- Presentation of ECAR (or lactate production) in Figure S3 would be desirable.
- Statistics in several experiments is unclear and not convincing: In Figure 2B, Is there a

significant difference between ID8Meso and ID8Meso+FRbeta+ TAM? Along the same line, the statistics on Figure 3D (IL10 production) should be clarified. And is it possible to statistically compare the data presented in Figure 4D (folr2 wt versus folr2-/-)? A similar comment can be made on the data on Figure 5C, that might benefit from increasing n (this might also apply to the data shown in Figure 6).

- The authors should explain the rationale for cyclophosphamide preconditioning in the experimental procedures in Figures 5 and 6, and at least comment on the results in the absence of such preconditioning.

- The authors could show the comparison of optimal versus sublethal doses of Meso-specific CAR-T cells.

Reviewer #2 (Remarks to the Author):

This interesting and clearly written paper addresses the immunosuppressive tumor microenvironment that is a major challenge to the success of tumor-specific T-cells, particularly those targeting solid tumors. The folate receptor beta is expressed on a range of tumors and most interestingly on tumor-associated macrophages (TAMs) that infiltrate many tumors and inhibit tumor-specific T-cells.

Rodriguez-Garcia et al., show that FRb is expressed on an immunosuppressive subpopulation of TAMs, and that eliminating them, either in knockout mice or by targeting them with a CAR results in increased control of FRb-ve tumors, although not elimination. However in combination with a tumor-targeting CAR, the FRb.CAR is able to provide prolonged control of tumor, provided the tumor-specific CAR is preceded by the FRb.CAR, implying that the TME must be altered radically before the tumor-specific CAR can work

- Please comment on demonstration that PDL-1 is upregulated of FRbeta+TAMs in Fig 2E but downregulated in 2 of 4 in Fig 2C. In fact there is a lot of discordance between the 4 samples in up and downregulated genes in Fig2C

- Figure 3b legend should be added to clarify meaning of colors

- Line 228, for readers benefit, please mention that FOLR2 is the gene for FRβ2.

- Could you discuss whether FRbeta is expressed on myeloid cells that control autoimmunity and whether autoimmunity might be dysregulated by this CAR

Could you discuss other immunosuppressive components of the TME, like Tregs and MDSCs and whether they express FRb

In mice that got sequential FRb then tumor-specific CAR, did all the mice progress after 19 days. Do you have any idea why? Do you find other suppressive cells in the progressive TME? and can either of the two CAR T-cells be detected?

- Did you evaluate the in vivo anti-tumor of the FRb.CAR using a TAM-containing tumor that expresses FRb?

Reviewer #3 (Remarks to the Author):

This study showed that TAMs that express FRβ have a distinct transcriptomic and metabolic profile versus inflammatory monocytes and suppress T cell responses. CART-cell based targeting of FRβ+ TAMs in the TME resulted in expansion of endogenous tumor-specific CD8+ T-cells and a delay in tumor progression. These results establish the foundation for CART-cell-based targeting of a specific population of TAMs.

Major critiques

- 1) Is FR β expressed on other cells in the TME besides TAMs? This point is relevant to both mouse data and human TCGA data?
- 2) The data on metabolic profiles of FR β + TAMs is limited to mitochondrial respiration (OCR), and the effect is limited to augmented OCR with addition of FCCP, but no effect on basal mitochondrial respiration. Was there a difference in mitochondrial mass or depolarization in FR β + TAMs? In addition, Seahorse will provide data on glycolysis that should be included.
- 3) The experimental conditions in Fig. 3 are unclear. ID8 is the specificity control that should result in background proliferation and cytokine responses, and ID8.hMeso is the positive control that should stimulate proliferation and cytokine responses. Id8.hMeso should be added alone; combined with FR β - TAMs; and combined with FR β + TAMs. This design enables direct comparison of the suppressive effect of FR β + TAMs vs. FR β - TAMs and vs. no TAMs. However, the setup seems to involve co-culture of CART-cells with ID8.hMeso + both population of TAMs, and TAMs alone without ID8.hMeso. This design should be clarified.
- 4) Fig. 6 shows the benefit of sequential CART-cells to deplete FR β +TAMs followed by tumor antigen-specific CART-cells. However, there doesn't seem to be a statistically significant difference between sequential therapy vs. anti-FR β +TAM CART-cells alone (Fig. 6H). The experiment seems to be stopped at around day 60, and it's unclear about progression of tumor in the sequential group after this time point. Please clarify whether survival in the combination group was better than anti-FR β +TAM CART-cells alone. It's also unclear in the BLI curves whether sequential therapy is better than anti-FR β +TAM CART-cells alone (Fig. 6F and G).
- 5) Fig. S4 on OT-1 proliferation is unconvincing. There's high level of background proliferation with irrelevant peptide, and no statistics are included.
- 6) Human correlative data from TCGA: Y-axis should clarify that data are PFS, not overall survival. The difference in stage # disease is very small. Is there additional data on stage (e.g. 3A, 3B, 3C). The term "OC" is used: is this restricted to epithelial ovarian cancer (EOC)? If yes, is the FR β high and low groups matched with regard to histology (HGSOC vs. others) and basic demographics, such as age?

Minor critiques

- 1) There should be citation of B7H4 macrophages being immunosuppressive in the EOC TME.
- 2) Ly6C^{int}Ly6G⁺ cells are referred to as MDSC (should state PMN-MDSC). This population was increased in mFR β CAR-treated group (Fig. 5G) and here they're referred to as "granulocytes." This terminology is important if CAR cells directed against TAMs increased PMN-MDSC.

REVIEWER COMMENTS

Reviewer #1 (Remarks to the Author):

The manuscript by Rodríguez-García et al. addresses the use of CAR T-cell mediated depletion of tumor-associated macrophages (TAM) as a feasible alternative to limit tumor growth and enhance anti-tumor immunity. The authors have used several very well-defined experimental systems (mouse tumor models) and adequately designed cellular tools, demonstrate that depletion of FOLR2+ macrophages has an impact on tumor growth, favors tumor elimination, and might complement other CAR T-cell based strategies. As a whole, the manuscript is of interest, timely, nicely introduced, and derives from previous observations of the group on the applicability of FR β -specific CAR T cells for acute myeloid leukemia. A relevant novelty of the manuscript is the identification of a marker (FOLR2) that might be useful to identify and target TAM in a wide variety of tumors.

Most of the experiments in the manuscript are clearly and properly controlled, and the manuscript includes relevant and solid supplementary information that complements the main text. As a whole, the manuscript is solid and very informative. However, a number of issues could be addressed for the manuscript to gain robustness and to fully support the authors conclusions.

General comments:

- All throughout the manuscript, and when referring to previous reports, the authors should clearly indicate the species where the data has been generated.

We thank the reviewer for this suggestion. We have now gone revised the manuscript in order to clarify the species in which data has been generated.

- The results from the Kaplan-Meier studies in ovarian cancer patients are particularly relevant. The authors could extend these studies to other tumor types, and determine whether this is a general effect. As an example, and given the fact that FOLR2 expression appears to be enhanced by M-CSF, a similar type of analysis could be done on breast carcinoma and using TCGA and an alternative validation cohort (Metabric?).

As suggested by the reviewer, we have extended these analyses to include other cancers in order to see if the use of FR β expression as a predictor for poor prognosis could be extrapolated to other tumor types. Overall, although FR β ⁺ TAMs are found in a wide range of cancers, our findings suggest that their prognosis value might not be generalizable to all tumor types. Larger datasets might be required to validate this observation. In any case, there is a number of tumor types in which there is a correlation between FOLR2 expression and reduced survival.

Km plotter is a web tool that allows the assessment of the effect of the expression different genes on survival in different types of cancer. Sources for the database include GEO, EG, and TGCA. mRNA data sources are gene chip or RNA seq. The data presented in original **Fig. 7e** corresponds to gene chip data, and equivalent data is only available for breast, lung and gastric cancer. From those, high expression of FOLR2 is only prognostic of poor survival in gastric cancer. These data have been now included as **Fig. S10a**. As suggested by the reviewer, we also validated the breast cancer data by using the alternative cohort Metabric, with similar results (p value=0.904, data not shown).

In addition, RNAseq data is available for 21 different cancer types. From all those tumor types, high FOLR2 expression significantly correlated with poor prognosis in bladder carcinoma, esophageal squamous cell carcinoma, kidney renal clear cell carcinoma, lung squamous cell carcinoma, stomach adenocarcinoma, and thymoma. Similar trends were observed for other tumor types such as rectum adenocarcinoma, liver hepatocellular carcinoma, or esophageal adenocarcinoma, however, these differences did not reach statistical significance. These data have been now included as **Fig. S10b**.

It is important to notice that by using RNAseq data, analysis for specific cancer stages is not available, and therefore, significant differences in survival were not detected for ovarian cancer patients (we only observed significant differences at advanced stages of the disease, as shown in **Fig. 7e**).

In addition to new **Fig. S10**, the following sentence commenting on these results has been included in the results section: *“A more extensive analysis of the correlation of FOLR2 and survival including different tumor types was performed revealing significant prognostic value in several tumor types including gastric cancer, bladder carcinoma, esophageal squamous cell carcinoma, kidney renal clear cell carcinoma, lung squamous cell carcinoma, stomach adenocarcinoma, and thymoma (Fig. S10).”*

- A major issue that arises from the differences at the transcriptional and cytokine levels between both subsets is the possible contribution of IL-10 to some of the observed functional effects (in fact, IL10 gene expression appears to parallel that of FOLR2). Thus, can the authors assess whether the effect of FOLR2-specific CAR-T cells is dependent on IL-10, either using IL10 KO mice or, alternatively, using anti-IL10 blocking antibodies? If so, can anti-IL-10 replace FOLR2-specific CAR-T cells in the experimental system used in Figure 6? In this regard, determination of IL10 production in the experiment presented in Figure S4 would be appropriate.

Similar to what the reviewer points out, higher levels of mIL-10 in the supernatants of the immunosuppression assay depicted in original **Fig. 3d** led us to hypothesize that IL-10 might be responsible for immunosuppressive function of FR β ⁺ TAMs. For that reason, we performed a similar *in vitro* proliferation assay by co-culturing hMeso-specific CAR-T cells, target ID8.hMeso tumor cells, and FR β ⁺ TAMs in the presence of a mIL-10R blocking antibody or the appropriate isotype control. Antigen-specific CAR-T cell activity was assessed by measuring proliferation (T cell counts) and cytokine secretion (mIFN γ concentration). Results from these experiments confirmed that the presence of FR β ⁺ TAMs reduced proliferation and cytokine secretion of CAR T-cells, but IL-10 blockade did not rescue CAR-T cell functionality. These data suggest that IL-10 is not a major contributor to FR β ⁺ TAM-mediated immunosuppression and that other mechanisms might be involved.

These data have been included as **Fig. S5b** (proliferation) and **Fig. S5c** (mIFN- γ secretion), and the following sentence describing the results has been incorporated in the manuscript: *“However, blockade of IL-10 by antibodies did not rescue proliferation (Fig. S5b) and IFN- γ secretion (Fig. S5c) by CAR T-cells, suggesting that IL-10 is not a major contributor for immunosuppression and other mechanisms might be involved.”* Also, the original **Fig. 3d** has been now moved to **Fig. S5a**.

Unfortunately, supernatants of experiment depicted in **Fig. S5d-f** are no longer available for mIL-10 quantification, but we would predict similar results by using OT-1 T cells to those obtained with CAR T cells described above.

Specific comments:

1. Figure 1 should include the presence of FOLR2⁺ macrophages in the peritoneum at t=0.

We thank the reviewer for the suggestion. We have now modified **Fig. 1a** to include frequency of FR β ⁺ TAMs at day 0. The manuscript has also been revised accordingly: *“A small fraction of peritoneal macrophages, as defined by their CD45, CD11b and F4/80 expression (Fig. S1a), showed positive surface expression of FR β immediately (12.36 \pm 4.51%) or two weeks (18.70 \pm 3.47%) after tumor inoculation (Fig. 1a). When tumors were allowed to grow to the point of significant ascites formation (approximately eight weeks), the FR β ⁺ subset of TAMs was increased to 52.14 \pm 11.42% of the total macrophage population (Fig. 1a), demonstrating that this is not a static population but one that increases over time.”*

2. The authors have gathered a very informative list of DEGs between FOLR⁺ and FOLR⁻ macrophages, but the validation experiments have addressed a limited number of markers that distinguish pro- and anti-tumoral macrophages (in Figure 2). A more detailed bioinformatics analysis of this transcriptional information would enrich the manuscript, possibly also hypothesizing the factors that might be responsible for this transcriptomic differences. As an example, can the authors check whether the expression of MAF and/or MAFB is different between both macrophage subsets at the protein level? Along the same line, and given the bioenergetics data included in the manuscript, what is the level of HIF factors in both subsets? The differential expression of EglN3 suggests that this might be the case....

As suggested by the reviewer, we have now included additional genes such as MAF and HIFs in the validation experiments at the protein level.

According to our transcriptomic data, both transcription factors mentioned by the reviewer, MAF and MAFB, were significantly upregulated in FR β ⁺ TAMs (1.52 and 1.49-fold, respectively). Following the reviewer recommendation, we now aimed at assessing differential expression of MAF (c-Maf) transcription factor between FR β ⁺ and FR β ⁻ TAMs at the protein level, by flow cytometry. Consistent with our transcriptomic data, we found that c-Maf was significantly upregulated in the FR β ⁺ TAM population (1.25-fold). In fact, this is in agreement with our conclusions that FR β ⁺ TAMs are M2-like, since c-Maf is known to control the expression of M2-related genes. For instance, it promotes IL-10 expression while inhibits that of IL-12 (Cao et al., J Immunol 2002, <https://pubmed.ncbi.nlm.nih.gov/12421951/>; Cao et al., J Immunol 2005, <https://pubmed.ncbi.nlm.nih.gov/15749884/>), and it is also critical for metabolic reprogramming for M2-polarization and in regulating M2-like macrophage-mediated T-cell immunosuppression (Liu et al., J Clin Invest 2020, <https://pubmed.ncbi.nlm.nih.gov/31945018/>). Therefore, based on our data and on previous reports, it could be hypothesized that the transcription factor c-MAF is one of the factors responsible for transcriptomic differences between FR β ⁺ and FR β ⁻ TAMs.

We have now included transcriptomic data on c-MAF and MAFB in the heatmap on **Fig. 2c**, where they clustered together with M2-related genes, and c-MAF expression data at the protein level is now presented as **Fig. S3b**.

In addition, the reviewer also suggested to quantify levels of HIF factors in both populations at the protein level based on egl3 downregulation at the transcriptional level in the FR β ⁺ TAM population (0.52-fold). This is actually consistent with previous reports indicating expression of Egl3 in pro-inflammatory M1 (GM-CSF) macrophages (Escribese, J Immunol 2012, <https://pubmed.ncbi.nlm.nih.gov/22778395/>). Egl3 is a prolyl hydroxylase that, in normoxia, mediates the hydroxylation of target proteins such as HIF-1 α and HIF-2 α , which are then targeted for proteasomal degradation. Under hypoxic conditions, the hydroxylation reaction is attenuated allowing HIFs to escape degradation resulting in increased expression of hypoxia-inducible genes. Egl3 is the most important isozyme in limiting physiological activation of HIFs in hypoxia.

Following the reviewer recommendation, we assessed protein levels of HIF-1 α and HIF-2 α (epas1), which are main targets for egl3. Consistent with our transcriptomic data, levels of HIF-1 α were very similar between FR β ⁺ and FR β ⁻ TAMs at the protein level. By contrast, frequency of HIF-2 α (epas1) positive cells was significantly higher within FR β ⁻ TAMs as compared to FR β ⁺ TAMs (10.21% vs 3.26%, 0.319-fold). This was also consistent with our transcriptomic data (0.45-fold). Expression of HIF-2 α has been reported to have a critical role for in regulating proinflammatory cytokine expression at low O₂, which would be consistent with the M1-like phenotype of FR β ⁻ TAMs (Imtiyaz et al., J Clin Invest 2010, <https://pubmed.ncbi.nlm.nih.gov/20644254/>).

We have now included transcriptomic data on epas1/HIF-2 α in the heatmap on **Fig. 2c**, where it clustered together with M1-related genes. Expression of HIF-1 α and HIF-2 α has now been included as **Fig. S3c-d** and the following sentence has been included to comment on this “*Levels of hypoxia-inducible factors (HIFs) were also assessed at the protein level in both populations. Levels of HIF-1 α were similar in FR β ⁺ and FR β ⁻ TAMs, while frequency of HIF-2 α positive cells was significantly higher in FR β ⁻ TAMs (Fig. S3c-d)*”:

On the other hand, as suggested by the reviewer, a more detailed bioinformatic analysis with focus on transcription factors could be performed to generate new hypotheses. We agree. A bioinformatic analysis was performed in which we analyzed enrichment or downregulation of gene sets that contain transcription factor binding sites for specific transcription factors by using the Legacy database. We observed that all of the differentially expressed gene sets were downregulated in FR β ⁺ TAMs rather than upregulated, as compared to FR β ⁻ TAMs (see figure below).

Looking at the list of the 30 top downregulated genesets (FDR<0.1), we found that some of the transcription factors are represented more than once, indicating that these factors may be relevant for the transcriptomic differences between both TAM subsets. The transcription factor that showed up more times across downregulated genesets of target genes was RSRFC4 (4 genesets), also known as MEF2A (myocyte enhancer factor 2). Other transcription factors that appear in multiple gene sets are GATA1 (3 genesets), LEF1 (2 genesets), and AREB6/Zeb1 (2 genesets). Of those, Gata1 and Zeb1 are significantly downregulated themselves in FR β ⁺ TAMs in our RNAseq data (0.46-fold and 0.7-fold, respectively). However, functional analysis would be required to identify which of this transcription factors account for the transcriptomic

differences found between FRβ⁺ and FRβ⁻ TAMs, which is beyond the scope of our current work and is an area of planned investigation.

3. Presentation of ECAR (or lactate production) in Figure S3 would be desirable.

We thank the reviewer for the suggestion; we have now included data on the extracellular acidification rates (ECAR) in **Fig. S4f**.

ECAR is an informative, measurable, surrogate of glycolytic metabolism which occurs in the cytoplasm. We observed no statistically significant differences in ECAR levels of FRβ⁻ and FRβ⁺ TAMs. We conclude that a metabolic attribute unique to FRβ⁺ TAMs is their enhanced energy reserve, conferred by their superior ability to consume oxygen at high rates. We have included the following statement in the manuscript to comment on these results: *“Finally, the extracellular acidification rate (ECAR) was measured to investigate the glycolytic capacity of FRβ⁻ and FRβ⁺ TAMs. As seen in Fig. S4f, no statistically significant differences in ECAR levels were reported. In aggregate, our findings imply that the enhanced metabolic capacity of FRβ⁺ TAMs is attributed to an increased energy reserve which is supported by an enhanced ability to consume oxygen at high rates under stimulatory conditions.”*

4. Statistics in several experiments is unclear and not convincing: In Figure 3B, Is there a significant difference between ID8Meso and ID8Meso+FRbeta+ TAM?

In **Fig. 3b**, although there is a trend for increased frequency of divided cells in “ID8.hMeso + FRβ⁺ TAM” as compared to “ID8.hMeso”, this difference is not statistically significant according to a one-way ANOVA test with Tukey’s multiple comparisons test (p value=0.3098). However, it is important to point out that there is a significant difference between “ID8.hMeso + FRβ⁺ TAM” and “ID8.hMeso + FRβ⁻ TAM” groups (p=0.04). In addition, when looking at mIFN-γ secretion (**Fig. 3c**), statistically significant differences were found between “ID8.hMeso + FRβ⁺ TAM” and “ID8.hMeso” (p=0.0144) as well as between “ID8.hMeso + FRβ⁺ TAM” and “ID8.hMeso + FRβ⁻ TAM” groups (p=0.0099), supporting our conclusion of that FRβ⁺ TAM inhibit CAR T function, whereas FRβ⁻ TAM do not have a discernable effect on CAR T cell activity.

In addition, in order to clarify statistics for this figure and also **Fig. 3c**, we have now also added significant differences with parental ID8 specificity control

Along the same line, the statistics on Figure 3D (IL10 production) should be clarified.

We thank the reviewer for this comment. Regarding **Fig. 3d**, we identified a minor error as the statistics that were shown in the original figure were calculated by a t-test instead of one-way ANOVA as it was stated in the figure legend and as it was performed for the other panels in this same figure. Differences in IL-10 levels between “CAR T cells + ID8.hMeso + FRβ⁺ TAM” and “CAR T cells + ID8.hMeso + FRβ⁻ TAM” do not reach statistical significance according to this test (p value = 0.0565).

We have updated statistics for this figure, modified the manuscript text to remove the word “significantly increased” and this panel to **Fig. S4a**.

And is it possible to statistically compare the data presented in Figure 4D (*folr2* wt versus *folr2*^{-/-})?

We sought to determine statistical significance across the time of cell culture, per reviewer questioning. In order to perform a statistical analysis to compare *folr2* wt versus *folr2*^{-/-} data, we pooled normalized cell index data obtained in both target cell types in a single graph.

A stark difference between *folr2* wt versus *folr2*^{-/-} cell killing is evident as all *folr2* wt cells are readily killed by mFRβ CAR T-cells, while *folr2*^{-/-} cells progressively grow. Statistically significant differences are observed as soon as the second time point (20 minutes after CAR T-cell addition) (p=0.0016), and are maintained throughout the experiment (p<0.0001) according to a two-way ANOVA with Sidak’s multiple comparison test.

A similar comment can be made on the data on Figure 5C, that might benefit from increasing n (this might also apply to the data shown in Figure 6).

The reviewer points out that the experiment depicted in **Fig. 5c** would benefit of a bigger n. We agree. This particular experiment has been conducted at least three times with reproducible results, although representative results for one of these experiments were shown in the original manuscript. We have now pooled bioluminescence and survival data for the three different experiments and updated the graphs of **Fig. 5c** and **5d** (n=13).

In the case of **Fig. 6**, during the course of this manuscript revision, and in order to confirm reproducibility of these results and to increase the n, we conducted an additional study combining the experiments outlined on **Fig. 6a** and **6d**. Trends in this new experiment were consistent with the previous studies, with sequential combination treatment being the only treatment that resulted in sustained antitumor effect. Because of this new experiment has a bigger n and therefore stronger statistics, and allowed us to directly compare the simultaneous versus the sequential combination treatment groups in the same study, as opposed of two independent studies as there were shown in the original manuscript, we have replaced **Fig. 6** with the more robust results from the new experiment.

5. The authors should explain the rationale for cyclophosphamide preconditioning in the experimental procedures in Figures 5 and 6, and at least comment on the results in the absence of such preconditioning.

We thank the reviewer for the observation. Preconditioning regimens with cyclophosphamide and IL-2 have been widely used in order to improve engraftment, persistence, and functional activity of adoptively transferred T-cells in both preclinical mouse models and clinical trials (Brentjens et al., Blood 2011, <https://pubmed.ncbi.nlm.nih.gov/21849486/>; Cheadle et al., Br J Haematol, <https://pubmed.ncbi.nlm.nih.gov/18477047/>; Cheadle et al., J Immunol 2014,

<https://pubmed.ncbi.nlm.nih.gov/24623129/>; Chinnasamy et al., J Clin Invest 2010, <https://pubmed.ncbi.nlm.nih.gov/20978347/>; Gattinoni et al., J Exp Med 2005, <https://pubmed.ncbi.nlm.nih.gov/16203864/>; Kakarla et al., Mol Ther 2013, <https://pubmed.ncbi.nlm.nih.gov/23732988/>). A statement clarifying this point has been included in the revised manuscript.

In the specific context of our mFR β CAR, antitumor effect has been compared to that of hCD19 CAR in the setting of Cy + IL2 preconditioning versus in the absence of preconditioning. While in the context of preconditioning mFR β CAR was able to limit tumor progression and prolong survival; in non-conditioned mice, treatment with mFR β CAR T cells did not have an impact on tumor progression or survival. Therefore, we conclude that antitumor efficacy of mFR β CAR T cells is dependent on Cy + IL2 preconditioning regimen.

Graphs showing these results have now been included in the manuscript as **Fig. S7c-d**, as well as the statement: “*Of note, antitumor efficacy of mFR β CAR T-cells was dependent on Cy+IL-2 preconditioning regimen (Fig. S7c-d).*”

6. The authors could show the comparison of optimal versus suboptimal doses of Meso-specific CAR-T cells.

We concur. In order to better reveal a potential therapeutic window in the combination experiments, we decided to use a suboptimal hMeso CAR T cell dose. Therefore, a pilot study comparing a high (8e6 CAR+T cells) and a low (4e6 CAR+T cells) dose of hMeso CAR T cells was performed in order to choose the appropriate dose to be applied in the combination study.

Results of the pilot study are shown now as **Fig. S8c**, depicting tumor progression based on bioluminescence imaging measurement. Of note, the UTD group BLI curve stops at day 12 post-treatment because of rapid tumor progression with just one mouse left in that group after that time point. In any case, the higher hMeso CAR T cell dose had a statistically significant better antitumor efficacy as compared to the lower dose, which did not appear to impact tumor progression. Therefore, 4e6 CAR+T cells was chosen as the suboptimal dose to create a therapeutic window for improvement upon combination with mFR β CAR T cells.

Reviewer #2 (Remarks to the Author):

This interesting and clearly written paper addresses the immunosuppressive tumor microenvironment that is a major challenge to the success of tumor-specific T-cells, particularly those targeting solid tumors. The folate receptor beta is expressed on a range of tumors and most interestingly on tumor-associated macrophages (TAMs) that infiltrate many tumors and inhibit tumor-specific T-cells.

Rodriguez-Garcia et al., show that FRb is expressed on an immunosuppressive subpopulation of TAMs, and that eliminating them, either in knockout mice or by targeting them with a CAR results in increased control of FRb-ve tumors, although not elimination. However in combination with a tumor-targeting CAR, the FRb.CAR is able to provide prolonged control of tumor, provided the tumor-specific CAR is preceded by the FRb.CAR, implying that the TME must be altered radically before the tumor-specific CAR can work

1. Please comment on demonstration that PDL-1 is upregulated on FRbeta+TAMs in Fig 2E but downregulated in 2 of 4 in Fig 2C. In fact, there is a lot of discordance between the 4 samples in up and downregulated genes in Fig2C.

We agree with the reviewer that, in general, there is variability between the 4 different samples in transcriptomic analysis. To clarify this point, the samples were obtained from 4 different mice, and from each mouse, two samples (one corresponding to FR β ⁺ TAMs, and one corresponding to FR β ⁻ TAMs) were obtained by flow-sorting. Subsequent analysis of RNA seq data revealed an inter-mouse variability, and for that reason, the statistical analysis of differential expression data was performed by using a paired analysis (thus, pairing each mouse FR β ⁻ and FR β ⁺ TAM sample). The results of this analysis revealed a statistically significant upregulation of PD-L1 on FR β ⁺ TAMs (1.29-fold, p=0.00754).

Of note, we chose to show individual data for each mouse on the heatmap on **Fig. 2c**. to acknowledge the inter-mouse variability and for full transparency. However, the way that this data should be analyzed is by comparing the color code between the FR β ⁻ and FR β ⁺ sample for each individual mouse (mouse #1 FR β ⁻ to mouse #1 FR β ⁺, mouse #2 FR β ⁻ to mouse #2 FR β ⁺, etc.), rather than looking at blue or red color in an individual sample to determine that one particular gene is up or down-regulated. For instance, in the case of PD-L1, when looking at the following simplified chart (obtained from **Fig. 2C data**) one observes that PD-L1 is upregulated on FR β ⁺ TAMs in 3 out of the 4 samples (#1, #2 and #3):

Also, these transcriptomic results were validated at the protein level by assessing PD-L1 expression by flow cytometry (**Fig. 2E**). In all of the samples tested (from 4 independent mice), expression of PD-L1 protein was higher on FR β ⁺ TAMs.

2. Figure 3b legend should be added to clarify meaning of colors

We thank the reviewer for the observation. We have now included a color-coded legend to clarify this point.

3. Line 228, for readers benefit, please mention that FOLR2 is the gene for FR β 2.

We thank the reviewer for the observation. We have now clarified this point.

4. Could you discuss whether FRbeta is expressed on myeloid cells that control autoimmunity and whether autoimmunity might be dysregulated by this CAR

Activated macrophages play a key role in several inflammatory diseases including atherosclerosis, lupus, psoriasis, rheumatoid arthritis, or ulcerative colitis. Activated macrophages in this disease context upregulate FR β , which is not found on quiescent macrophages or other normal cells, and therefore, this receptor has been used to selectively deliver imaging and therapeutic agents utilizing folate as a targeting molecule (Elo et al., J Neuroinflammation 2019,

<https://pubmed.ncbi.nlm.nih.gov/31796042/>; Nakashima-Matsushita et al., Arthritis Rheum 1999, <https://pubmed.ncbi.nlm.nih.gov/10446858/>; Turk et al., Arthritis Rheum 2002, <https://pubmed.ncbi.nlm.nih.gov/12124880/>; van der Heijden et al., Arthritis Rheum 2009, <https://pubmed.ncbi.nlm.nih.gov/19116913/>; Xia et al., Blood 2009, <https://pubmed.ncbi.nlm.nih.gov/18952896/>). Also, the use of agents that reduce this population of activated macrophages ameliorated the symptoms on animal models of several of these autoimmune diseases, demonstrating therapeutic benefit of targeting FR β ⁺ macrophages (Feng et al., Arthritis Res Ther 2011, <https://pubmed.ncbi.nlm.nih.gov/21477314/>; Furusho et al., J Am Heart Assoc 2012, <https://pubmed.ncbi.nlm.nih.gov/23130174/>; Hu et al., Arthritis Res Ther 2019, <https://pubmed.ncbi.nlm.nih.gov/31174578/>; Low et al., Acc Chem Res 2008, <https://pubmed.ncbi.nlm.nih.gov/17655275/>; Shen et al., Mol Pharm 2008, <https://pubmed.ncbi.nlm.nih.gov/23641923/>; Varghese et al., Mol Pharm 2007, <https://pubmed.ncbi.nlm.nih.gov/17848087/>).

We believe that in the setting of a cancer patient without underlying autoimmune diseases (a common exclusion criteria for CAR T cell trials), FR β CAR T cells would not have a negative impact on regulation of autoimmunity, as FR β is only expressed on alternatively activated macrophages. In the case of a cancer patient with any of the above mentioned autoimmune diseases, the use of FR β CAR T cells could be reconsidered, although targeting FR β ⁺ macrophages could be of benefit for both conditions based on our results or those previously reported (see above).

To comment on this issue, the following sentence has been included in the Discussion section: “*Another concern might be the expression of FR β in activated macrophages in several autoimmune diseases such as rheumatoid arthritis. The use of agents that reduce this population of activated macrophages ameliorated symptoms in animal models of inflammatory disease demonstrating a therapeutic benefit of targeting FR β ⁺ macrophages also in the setting of inflammatory diseases without deregulating autoimmunity.*”

5. Could you discuss other immunosuppressive components of the TME, like Tregs and MDSCs and whether they express FR β

We agree with the reviewer that it is of great relevance to assess expression of FR β in other components of the TME in order to provide evidence of that depletion of FR β ⁺ TAMs by CAR T cells is the main mechanism of antitumor effect.

A previous study that analyzed FR β expression in 992 human tumor sections including 20 tumor types concluded that FR β was more pronounced in cells within the stroma, and implicated macrophages as being the main cell population in the TME that expresses FR β (Shen et al., Oncotarget 2015, <https://pubmed.ncbi.nlm.nih.gov/25909292/>).

We had included FR β expression data on MDSCs in the original manuscript. We have now extended this data into monocytic MDSCs (M-MDSCs, gated as live/CD45+CD11b+F4/80-Ly6C+Ly6G-) and polymorphonuclear MDSCs (PMN-MDSCs, gated as live/CD45+CD11b+F4/80-Ly6C+Ly6G+) (gating strategies for both populations are shown in **Fig. S2c**). As observed in **Fig. S2e**, none of these MDSCs populations express significant levels of FR β .

We also assessed FR β expression on regulatory T cells (Tregs). FR β is a myeloid lineage marker and therefore not expected to be expressed on Tregs. Tregs (gated as live/CD45+CD3+CD4+CD25+FoxP3+) were detected in the ascites of ID8-bearing mice and stained for FR β expression by flow cytometry. As anticipated, we did not detect significant levels of FR β expression on Tregs. In fact, there are many reports in the literature demonstrating expression of the folate receptor 4 (FR4) on Tregs, but expression of FR β has not been reported (Jia et al., Immunol Invest 2009, <https://pubmed.ncbi.nlm.nih.gov/19860584/>; Yamaguchi et al., Immunity, <https://pubmed.ncbi.nlm.nih.gov/17613255/>).

This data is now included in the revised manuscript as **Fig. S2d** (gating strategy) and **Fig. S2e** (representative histogram and frequency of FR β positive cells).

6. In mice that got sequential FR β then tumor-specific CAR, did all the mice progress after 19 days. Do you have any idea why? Do you find other suppressive cells in the progressive TME? and can either of the two CAR T-cells be detected?

To clarify, in **Fig. 6f**, we chose to show BLI data from the sequential combination treatment only up to 19 days post treatment so that it was comparable to the study shown in **Fig. 6b** corresponding to simultaneous combination treatment, for which day 19 is the last day for which BLI data was acquired. In fact, BLI data for the study depicted in **Fig. 6e-h** was collected up to day 38 for all groups (see graph below), and up to day 53 for the sequential combination group. As can be seen, control of tumor growth was maintained until the last time point, when tumors in 2 out of 5 mice were still regressing. Please see in the graph below complete BLI data for the study outlined on **Fig. 6e**:

During the course of this manuscript revision, we conducted an additional study combining the experiments outlined on **Fig. 6a** and **6d**. These experiments directly compare the simultaneous combination treatment versus the sequential combination treatment in the same experiment, and including hCD19, mFRβ and hMeso CAR T cells alone as controls. Trends in this new experiment were consistent with the previous study, with sequential combination treatment being the only treatment that resulted in sustained antitumor effect and prolonged survival as compared to control groups. In addition, BLI data was collected until day 48 for all the groups (when most animals in the control groups had to be euthanized because of high tumor burden), and up to day 69 for the sequential combination treatment group. Remarkably, all of the tumors at this time point were in complete remission by BLI measurement ($-94.69 \pm 6.44\%$ of percentage of tumor growth versus day -1). Data of this new experiment has now been included in **Fig. 6**, replacing that of the original figure.

As the reviewer may observe, the amount of tumor regressions and overall antitumor effect of the new experiment is greater than in the previous experiment. This could be due to smaller tumor burden at the moment of treatment of the last experiment (BLI=2.89e8) as compared to the previous experiment (BLI=7.18e8), although in both experiments mice were treated 21 days after tumor inoculation. This might indicate that tumor burden could be a factor influencing antitumor response.

In this new experiment, peritoneal cells were collected from representative mice at the endpoint (around D56 for hMeso and mFRβ+hMeso groups, and D69 for mFRβ→hMeso group), and analyzed for the presence of suppressive cell populations in the TME by flow cytometry, as suggested by the reviewer. Frequencies of FRβ+ TAMs and MDSCs were lower in mice treated with the sequential combination group, even at this late time point, as compared to hMeso and mFRβ+hMeso simultaneous combination groups. In addition, although not statistically significant, trends to lower frequencies of Tregs were also found. These results are consistent with the low to undetectable tumor burden in mice treated with the sequential combination at this time point, and has now been included as **Fig. S8f-h**.

In addition, to respond to the question by the reviewer of whether either of the CAR T-cells could be detected after infusion and to better understand the mechanisms of improved antitumor effect in the sequential combination treatment group, in this new study, hMeso CAR T-cells were generated from CD45.1 B6 mice, in order to be able to track tumor-specific CAR T-cells (CD45.1+GFP+) and differentiate them from TAM-specific CAR (CD45.2+GFP+) and endogenous T-cells (CD45.2+GFP-). In fact, low but detectable frequencies of both CAR T-cell populations were still found in the TME at the endpoint (d77), indicating that these CAR T-cells are able to persist for a long period of time in the TME in the regressing lesions. This data has now been included as **Fig. S8d-e**.

In addition, we were able to detect increased concentrations of hMeso CAR T-cells 9 days after administration in the blood of mice that had been previously treated with mFRβ CAR T-cells (sequential treatment), possibly explaining the

increased early antitumor effect, suggesting a better expansion of the CAR T-cells in the absence of immunosuppressive TAMs. We have included this data as **Fig. 6e**.

Finally, another significant finding of these new experiments that may contribute to understanding the mechanism of the improved antitumor effect of the sequential combination treatment is the induction of a more robust endogenous T-cell response, as demonstrated by a higher concentration of recipient CD45.2⁺ T cells in the blood 9 days after treatment. This data has been now included as **Fig. 6f**. Nevertheless, this expansion in the blood was contracted at day 21 post-treatment, as shown in the graph below:

Fig. 6 has been replaced by the new data, and the text in the manuscript has been updated accordingly.

7. Did you evaluate the *in vivo* anti-tumor of the FRb.CAR using a TAM-containing tumor that expresses FRb?

Yes, we had evaluated the efficacy of mFR β CAR T cells in mice inoculated with ID8 tumor cells that had been genetically engineered to express mFR β (ID8.mFR β RFP fLuc) to confirm the functionality of the CAR *in vivo*. Results of this experiment are shown in **Fig. S6i** of the original manuscript. In this model, both, tumor cells and recruited TAMs express FR β , and the antitumor efficacy observed might be attributable to a combination of targeting both cell populations.

Reviewer #3 (Remarks to the Author):

This study showed that TAMs that express FR β have a distinct transcriptomic and metabolic profile versus inflammatory monocytes and suppress T cell responses. CART-cell based targeting of FR β ⁺ TAMs in the TME resulted in expansion of endogenous tumor-specific CD8⁺ T-cells and a delay in tumor progression. These results establish the foundation for CART-cell-based targeting of a specific population of TAMs.

Major critiques

1) Is FR β expressed on other cells in the TME besides TAMs? This point is relevant to both mouse data and human TCGA data?

We thank the reviewer for the comment. Please, see response to reviewer #2 point 5 and **Fig. S2**. We did not find expression of FR β in Tregs, MDSCs (monocytic or polymorphonuclear) or tumor cells in the TME.

2) The data on metabolic profiles of FR β ⁺ TAMs is limited to mitochondrial respiration (OCR), and the effect is limited to augmented OCR with addition of FCCP, but no effect on basal mitochondrial respiration. Was there a difference in mitochondrial mass or depolarization in FR β ⁺ TAMs? In addition, Seahorse will provide data on glycolysis that should be included.

As the reviewer observed, baseline oxygen consumption rates (OCR) were similar in FR β ⁻ and FR β ⁺ TAMs, and the oxidative features of FR β ⁺ TAMs were only significantly increased under depolarizing conditions, consistent with an increased energy generating capacity.

We agree with the reviewer that mitochondrial mass as well as mitochondrial membrane potential (MMP) are important determinants of metabolic capacity. To provide increased insight into the metabolic features of FR β ⁻ vs FR β ⁺ TAMs, we measured mitochondrial mass by flow cytometry, by using MitoTracker Green FM (a membrane-permeable dye commonly used to stain mitochondrial mass). We observed similar levels of MitoTracker FM staining in FR β ⁻ and FR β ⁺ TAMs, consistent with the baseline OCR profile. The MMP reflects the separation of electrical charge across the inner mitochondrial membrane. Several reports show how the high MMP levels induce reactive oxidative stress and trigger a senescent phenotype. Using tetramethylrhodamine, methyl ester (TMRM), we provide evidence that the MMP is similar in FR β ⁻ and FR β ⁺ TAMs. Finally, metabolism is not limited to the mitochondria. Extracellular acidification rate (ECAR) is an informative, measurable, surrogate of glycolytic metabolism which occurs in the cytoplasm. We observed no differences in ECAR levels of FR β ⁻ and FR β ⁺ TAMs. Overall, we conclude that a metabolic attribute unique to FR β ⁺ TAMs is their enhanced energy reserve, conferred by their superior ability to consume oxygen at high rates.

We have now included mitochondrial mass, MMP, and ECAR data as **Fig. S4d**, **Fig. S4e**, and **Fig. S4f**, respectively, and modified the manuscript to include the following statements:

*“M1 and M2-polarized macrophages also display unique metabolic properties, with M1 macrophages utilizing glycolysis to generate ATP, and M2 macrophages obtaining much of their energy from oxidative phosphorylation. FR β ⁻ and FR β ⁺ TAMs were harvested and directly sorted to characterize their metabolic activity using a standard Seahorse assay. Oxygen consumption rates (OCR) were similar in FR β ⁻ and FR β ⁺ TAMs at baseline, but under depolarizing conditions, the oxidative features of FR β ⁺ TAMs were significantly higher (**Fig. 2h**), consistent with an increased energy generating capacity. Of note, these higher maximal OCR levels are consistent with an M2 phenotype of oxidative metabolism (**Fig. 2h and Fig. S4a**). When FR β ⁺ TAMs were cultured in the presence of LPS, maximum OCR was reduced to similar levels as M1-like FR β ⁻ TAMs, which were unaffected by LPS addition (**Fig. S4b**), while the addition of mIL-4, typically used to polarize macrophages towards a M2 phenotype, had the opposite effect, increasing maximum OCR in both, FR β ⁻ and FR β ⁺ TAM populations (**Fig. S4c**). To provide increased insight into the metabolic features of FR β ⁻ and FR β ⁺ TAMs, we measured mitochondrial mass and mitochondrial membrane potential (MMP) by flow cytometry by using MitoTracker FM and tetramethylrhodamine, methyl ester (TMRM), respectively. Similar levels were observed for both parameters in FR β ⁻ and FR β ⁺ TAMs (**Fig. S4d-e**), consistent with the baseline OCR profile. Finally, the extracellular acidification rate (ECAR) was measured to investigate the glycolytic capacity of FR β ⁻ and FR β ⁺ TAMs. As seen in **Fig. S4f**, no statistically significant differences in ECAR levels were reported. In aggregate, our findings imply that the enhanced metabolic*

capacity of FR β ⁺ TAMs is attributed to an increased energy reserve which is supported by an enhanced ability to consume oxygen at high rates under stimulatory conditions.”

3) The experimental conditions in Fig. 3 are unclear. ID8 is the specificity control that should result in background proliferation and cytokine responses, and ID8.hMeso is the positive control that should stimulate proliferation and cytokine responses. ID8.hMeso should be added alone; combined with FR β ⁻ TAMs; and combined with FR β ⁺ TAMs. This design enables direct comparison of the suppressive effect of FR β ⁺ TAMs vs. FR β ⁻ TAMs and vs. no TAMs. However, the setup seems to involve co-culture of CART-cells with ID8.hMeso + both population of TAMs, and TAMs alone without ID8.hMeso. This design should be clarified.

We thank the reviewer for the observation. To clarify the design of the experiment shown in **Fig. 3**; all of the groups contain hMeso-specific CAR T cells. As the reviewer points out, ID8 is the specificity control for the CAR (grey bar) and ID8.hMeso is the positive control for proliferation and cytokine production in the absence of TAMs (blue bar). In order to directly compare suppressive effects of FR β ⁺ TAMs and FR β ⁻ TAMs, experimental groups were included containing CAR T cells, ID8.hMeso tumor cells and either FR β ⁻ (black bar) or FR β ⁺ (red bar) TAMs. No groups are included involving TAMs alone without ID8.hMeso.

4) Fig. 6 shows the benefit of sequential CART-cells to deplete FR β ⁺TAMs followed by tumor antigen-specific CART-cells. However, there doesn't seem to be a statistically significant difference between sequential therapy vs. anti-FR β ⁺TAM CART-cells alone (Fig. 6H). The experiment seems to be stopped at around day 60, and it's unclear about progression of tumor in the sequential group after this time point. Please clarify whether survival in the combination group was better than anti-FR β ⁺TAM CART-cells alone. It's also unclear in the BLI curves whether sequential therapy is better than anti-FR β ⁺TAM CART-cells alone (Fig. 6F and G).

As the reviewer points out, in this particular study, survival is not significantly different between sequential therapy and mFR β CAR alone, although mean survival is extended from 45 to 57 days. However, it is important to point out that the sequential treatment group is the only group with significantly prolonged survival as compared to control hCD19 CAR or hMeso CAR alone, while mFR β CAR alone did not significantly improve mice survival at this dose. Similarly, in original **Fig. 6g**, statistical significance was not reached between sequential therapy and mFR β CAR alone, probably because of the high intra-group variability, although again, sequential therapy was significantly better than hCD19 CAR or hMeso CAR alone.

As of BLI graph depicted on the original **Fig. 6f**, we chose to show BLI data only up to day 19 post-treatment so that it could be directly comparable to the graph shown in **Fig. 6b-c**, depicting antitumor activity of the simultaneous combination treatment. Up to this time point, the sequential treatment is significantly better as compared to mFR β CAR alone at day 17 after treatment. However, BLI data for this study was collected up to day 38 for all groups. When looking at this extended data, significant differences are observed between mFR β CAR alone and sequential combination group at two additional timepoints; day 23 and 27 post-treatment. In addition, BLI data was collected up to day 53 for the sequential combination group, when tumors in 2 out of 5 mice were still regressing. Please, see in the graph below complete BLI data for the study outlined on original **Fig. 6e**:

Beyond day 53 post-treatment, only body weight data was collected in order to monitor the formation of ascites up to day 63, when 2 out of the 5 mice in the sequential combination treatment group had not reached a 10% of body weight gain, which is established as the endpoint for the survival curves. In spite of this, all the remaining animals were terminated at this time point.

During the course of this manuscript revision, and in order to determine if these results were reproducible and to increase the n, we conducted an additional study combining the experiments outlined on **Fig. 6a** and **6d**. This is, directly comparing the simultaneous combination treatment versus the sequential combination treatment in the same experiment, and including hCD19, mFR β and hMeso CAR T cells alone as controls. Trends in this new experiment were consistent with the previous studies, with sequential combination treatment being the only treatment that resulted in sustained antitumor effect. In addition, BLI data for all groups was collected until day 48, when most animals in the control groups had to be euthanized because of high tumor burden. Up to this time point, the sequential combination treatment group was significantly better as compared to all the other groups at several time points (days 2, 5 and 48 vs hCD19 (*); day 48 vs hMeso (ζ); days 20 and 34 vs mFR β (&); and days 2, 5, 20, 27, 34 and 48 vs simultaneous combination (#)). In addition, mice in the sequential combination treatment group were continued to be imaged up to day 69. Survival curves were also generated for this new *in vivo* study by considering a body weight gain of 10% as the endpoint, indicative of ascites formation. In this new experiment, survival of the mice from the sequential combination group was significantly longer as compared to all the other groups (including mFR β CAR T cells alone). Remarkably, all of the tumors in the sequential combination group remained in remission by BLI measurement at the end of the study at day 69 post-treatment (-94.69 \pm 6.44% of percentage of tumor growth versus day -1):

Because of this new experiment has a bigger n and therefore stronger statistics, and allow us to directly compare the simultaneous versus the sequential combination treatment groups in the same study and at longer time points as opposed of two separated studies as there were shown in the original manuscript, we have replaced **Fig. 6** with this new experiment, and modified the manuscript accordingly. We also now include data on concentration of hMeso CAR T cells in the blood 9 days after their administration (**Fig. 6e**) as well as concentration of endogenous T-cells at that same timepoint (**Fig. 6f**).

5) Fig. S4 on OT-1 proliferation is unconvincing. There's high level of background proliferation with irrelevant peptide, and no statistics are included.

We agree with the reviewer that high levels of background and intra-group variability make proliferation results unconvincing as statistical differences were not detected. A possible reason for this is that two different experiments were pooled in a single graph. We have now made a new figure based on a single experiment, representative of two, and we have updated statistics.

As the reviewer points out, there is a certain level of background proliferation with irrelevant peptide, due to high variability between replicates in this particular condition. In any case, the addition of FR β ⁺ TAMs to the coculture resulted in significantly decreased proliferation as compared to the positive control, OT-1 + CSC, and importantly, as compared to the addition of FR β ⁻ TAMs based on an unpaired t-test. In addition, the analysis of mIFN- γ in the supernatants of this same experiment show a very clean result for the irrelevant peptide group, which was significantly lower than in all the other groups. Also, significant differences between the group with FR β ⁺ TAMs and all the other groups support the results of that this population of TAMs significantly affect the functionality of antigen-specific T cells (OT-1 T cells in this case). This updated data is now shown in **Fig. S4d-f**.

6) Human correlative data from TCGA: Y-axis should clarify that data are PFS, not overall survival.

We thank the reviewer for the comment. Although it was already stated in the figure legend and in the text, we have now also changed the axis to clarify that it corresponds to progression-free survival (PFS) and not overall survival (OS).

The difference in stage # disease is very small. Is there additional data on stage (e.g. 3A, 3B, 3C).

We agree with the reviewer that, although statistically significant, differences on PFS in stage 3 are small. Unfortunately, the Km plotter tool does not include additional data on stage. However, there is additional data on cancer grade. For instance, for EOC stage 3 grades 2 and 3 there was data from enough patients to plot survival curves. In both cases high expression of FR β correlated with worse PFS (see graphs below).

The term “OC” is used: is this restricted to epithelial ovarian cancer (EOC)? If yes, is the FRβ high and low groups matched with regard to histology (HGSO vs. others) and basic demographics, such as age?

The term “OC” includes serous and endometrioid histologies of epithelial ovarian cancer. We have relabeled the graphs to say “EOC” instead of “OC”.

If we look at the prognosis data by histology, the trends are the same than for the analysis including both. However, differences were statistically significant only in the serous histology, most likely due to the low number of patients in the endometrioid histology (please, see graphs below).

Unfortunately, the Km plotter tool does not provide data on basic demographics such as age.

Minor critiques

1) There should be citation of B7H4 macrophages being immunosuppressive in the EOC TME.

We thank the reviewer for the suggestion. In fact, our lab has previously developed a CAR against B7-H4 since this molecule is also expressed in ovarian cancer cells. However, besides inducing antitumor effect, this CAR induced delayed lethal toxicity, suggesting that this target might be expressed in some healthy tissues and therefore, targeting FRβ would be a safer approach (Smith et al., Mol Ther 2016, <https://pubmed.ncbi.nlm.nih.gov/27439899/>).

In any case, we have incorporated the following sentence in the discussion: “*These findings may also be generalizable to the use of alternative targets (Kryczek et al., J Exp Med 2006, <https://pubmed.ncbi.nlm.nih.gov/16606666/>) or alternative macrophage disrupting agents combined with other forms of T-cell provoking immunotherapies.*”

2) Ly6CintLy6G+ cells are referred to as MDSC (should state PMN-MDSC). This population was increased in mFRβ CAR-treated group (Fig. 5G) and here they’re referred to as “granulocytes.” This terminology is important if CART cells directed against TAMs increased PMN-MDSC.

We thank the reviewer for the observation, as we acknowledge the mistake of denominating MDSCs the PMN-MDSC subset, without taking into account M-MDSCs on **Fig. S2**. We now also include data on FR β expression on M-MDSCs (Ly6C^{high}Ly6G⁻). Therefore, PMN-MDSCs were gated as live/CD45⁺CD11b⁺F4/80⁻Ly6C^{int}Ly6G⁺, and M-MDSCs were defined as live/CD45⁺CD11b⁺F4/80⁻Ly6C⁺Ly6G⁻. It is important to highlight at this point that for defining each of the MDSC populations, F4/80 positive cells have been excluded, as this is a marker for mature macrophages which can be used to distinguish TAMs and M-MDSCs (Bronte et al., Nat Commun 2016, <https://pubmed.ncbi.nlm.nih.gov/27381735/>).

As pointed out by the reviewer, in **Fig. 5g** and **Fig. S8f** we used a similar gating strategy to identify granulocytes and inflammatory monocytes, except for the exclusion of F4/80 positive cells. The aim of this experiment was to identify myeloid cell populations newly expanded or recruited upon treatment with mFR β CAR T cells, as evidenced by an increase in CD11b⁺ cells both in the blood and at the tumor sites (ascites), as opposed to myeloid cells already present in the TME in non-treated mice (**Fig. S2**).

Now, one of the main challenges in the field is the distinction between neutrophils (granulocytes) or monocytes and PMN-MDSCs and M-MDSCs, respectively, since they are phenotypically identical. The main differential trait between these cell populations is the potent immunosuppressive activity of MDSCs, and therefore, isolation and functional characterization of the cells would be required. Classical activation of neutrophils and monocytes is characterized, among other things, by release of pro-inflammatory cytokines, while pathological activation (MDSC) results into the release of predominantly anti-inflammatory cytokines. In fact a study thoroughly comparing neutrophils to PMN-MDSCs (which are phenotypically identical), TNF- α secretion by neutrophils was found as one of the main distinctive characteristics (Youn et al., J Leukoc Biol 2012, <https://pubmed.ncbi.nlm.nih.gov/21954284/>).

Although we have not performed immunosuppression assays with these cells, we found that CD11b⁺ cells isolated from the altered TME after mFR β CAR T-cell treatment secreted higher levels of TNF- α and lower levels of IL-10 (**Fig. 5h**), suggesting that myeloid cell populations increased upon CAR T-cell treatment would correspond to inflammatory monocytes and neutrophils rather than MDSCs, prompting us to use a different terminology to name these populations.

Additionally, in order to provide more insight in the characterization of the monocyte cell populations infiltrating the TME after FR β ⁺ TAM depletion by CAR T-cells, we also looked at CX3CR1 and CCR2 expression. 6 days after CAR T-cell treatment, population of classical/inflammatory monocytes, defined as CX3CR1⁻CCR2⁺ was significantly increased as compared to control groups, while nonclassical/patrolling monocytes defined as CX3CR1⁺CCR2⁻ were significantly decreased. In fact, a previous study defining monocyte subsets in peripheral blood of mice described that the CX3CR1⁻CCR2⁺ population is actively recruited into inflamed tissues and then differentiate into dendritic cells which have the ability to stimulate naive T-cells, supporting our hypothesis that these cells contribute to the stimulation of an endogenous antitumor immune T-cell response (Geissmann et al., Immunity 2003, <https://pubmed.ncbi.nlm.nih.gov/12871640/>). Further supporting our results, previous reports exist indicating that TAM depletion results into a recruitment of inflammatory monocytes with a CX3CR1⁻ phenotype that contribute to the antitumor effect (Etzerodt et al., J Exp Med 2019, <https://pubmed.ncbi.nlm.nih.gov/31375534/>). This data has been now included as **Fig. S7h** and the sentence “*Consistently, further phenotypic analysis of infiltrating monocyte subsets revealed higher frequencies of classical/inflammatory monocytes (CX3CR1⁻CCR2⁺) while lower frequencies of nonclassical/patrolling monocytes (CX3CR1⁺CCR2⁻, Fig. S7h).*” has been inserted in the results section of the manuscript to comment on these results. Also the term “granulocytes” has been replaced by “neutrophils” through the manuscript, as it is more specific.

In addition to addressing all of the reviewers’ comments, we have now included additional data on antitumor efficacy of mFR β CAR T-cells in *Fcrl2*^{-/-} mice, in order to demonstrate antigen-specificity of the CAR *in vivo*. BLI and survival data are shown as **Fig. S7a**, showing no effect of the CAR T-cells on tumor progression or survival in the knockout mice.

REVIEWERS' COMMENTS

Reviewer #1 (Remarks to the Author):

The authors have adequately and very appropriately addressed all my questions and comments. While the original version was in itself, a solid and convincing work, including the new set of experiments have increased its clarity and significance. Thus, the revised version is improved, and I have no further comments or questions.

Reviewer #2 (Remarks to the Author):

The authors have very thoroughly and thoughtfully addressed the reviewers' comments and new experiments support their conclusions. The paper is novel, innovative and well-written

Reviewer #3 (Remarks to the Author):

The revised manuscript has been significantly strengthened by new data. This is particularly the case with the revised Fig. 6 showing survival benefit of sequential CART-cells to deplete FR β +TAMs followed by tumor antigen-specific CART cells. This statement isn't completely accurate: "hMeso CAR T-cells proliferated in the presence of ID8.hMeso target cells, however, this proliferation was significantly inhibited by FR β + TAMs, but not by FR β - TAMs (Fig. 3b)" since there's no clear visual or statistical difference in CAR-T cell proliferation between coculture with FR β + TAMs (red) versus no TAMs (blue). The IFN-g data (Fig. 3B) is convincing. My other critiques were addressed with the addition of new data and/or clarifications in the revision.

REVIEWERS' COMMENTS

Reviewer #1 (Remarks to the Author):

The authors have adequately and very appropriately addressed all my questions and comments. While the original version was in itself, a solid and convincing work, including the new set of experiments have increased its clarity and significance. Thus, the revised version is improved, and I have no further comments or questions.

We thank the reviewer for the positive assessment of our revised version of the manuscript.

Reviewer #2 (Remarks to the Author):

The authors have very thoroughly and thoughtfully addressed the reviewers' comments and new experiments support their conclusions. The paper is novel, innovative and well-written

We thank the reviewer for the positive assessment of our revision of the manuscript.

Reviewer #3 (Remarks to the Author):

The revised manuscript has been significantly strengthened by new data. This is particularly the case with the revised Fig. 6 showing survival benefit of sequential CART-cells to deplete FR β +TAMs followed by tumor antigen-specific CART cells.

This statement isn't completely accurate: "hMeso CAR T-cells proliferated in the presence of ID8.hMeso target cells, however, this proliferation was significantly inhibited by FR β + TAMs, but not by FR β - TAMs (Fig. 3b)" since there's no clear visual or statistical difference in CAR-T cell proliferation between coculture with FR β + TAMs (red) versus no TAMs (blue). The IFN-g data (Fig. 3B) is convincing. My other critiques were addressed with the addition of new data and/or clarifications in the revision.

We thank the reviewer for the positive evaluation of the revised manuscript.

We have now modified the statement on Fig. 3b so that it is completely accurate. The statement is as follows: "Proliferation of hMeso CAR T-cells was significantly reduced in the presence of FR β + TAMs as compared to the condition in which FR β -TAMs were added."